# Accelerating SGDM via Learning Rate and Batch Size Schedules: A Lyapunov-Based Analysis

**Yuichi Kondo**  *ce265022@meiji.ac.jp*
*Meiji University*

**Hideaki Iiduka**  *iiduka@cs.meiji.ac.jp*
*Meiji University*

**Reviewed on OpenReview:** *https://openreview.net/forum?id=s6DTv7Sorj*

## Abstract

We analyze the convergence behavior of stochastic gradient descent with momentum (SGDM) under dynamic learning-rate and batch-size schedules by introducing a novel and simpler Lyapunov function. We extend the existing theoretical framework to cover three practical scheduling strategies commonly used in deep learning: a constant batch size with a decaying learning rate, an increasing batch size with a decaying learning rate, and an increasing batch size with an increasing learning rate. Our results reveal a clear hierarchy in convergence: a constant batch size does not guarantee convergence of the expected gradient norm under our Lyapunov-based analysis, whereas an increasing batch size does, and simultaneously increasing both the batch size and learning rate achieves a provably faster decay. Empirical results validate our theory, showing that dynamically scheduled SGDM significantly outperforms its fixed-hyperparameter counterpart in convergence speed. We also evaluated a warmup schedule in experiments, which empirically outperformed all other strategies in convergence behavior.

## 1 Introduction

Stochastic gradient descent (SGD) (Robbins & Monro, 1951) and its variants are fundamental methods for training deep neural networks (DNNs) as they enable efficient optimization of empirical risk minimization problems. This paper focuses on a representative variant, SGD with Momentum (SGDM) (Polyak, 1964; Nesterov, 1983; Sutskever et al., 2013).

Typical SGDM algorithms include the Stochastic Heavy-Ball method (SHB) (Polyak, 1964) and its normalized variant (NSHB) (Gupal & Bazhenov, 1972), defined as follows:

$$\begin{cases} \text{SHB:} & \boldsymbol{m}_t = \beta \boldsymbol{m}_{t-1} + \nabla f_{B_t}(\boldsymbol{\theta}_t), \\ \text{NSHB:} & \boldsymbol{m}_t = \beta \boldsymbol{m}_{t-1} + (1-\beta)\nabla f_{B_t}(\boldsymbol{\theta}_t), \end{cases}$$

$$\boldsymbol{\theta}_{t+1} = \boldsymbol{\theta}_t - \lambda_t \boldsymbol{m}_t,$$

where $\lambda_t$ denotes the learning rate, $\beta \in [0,1)$ is the momentum coefficient, and $\nabla f_{B_t}(\boldsymbol{\theta}_t)$ is the stochastic gradient computed on the mini-batch $B_t$.

SGDM, including SHB and NSHB, generalizes the standard SGD. By leveraging historical gradient information, SGDM can accelerate convergence and stabilize the optimization process, as demonstrated in previous studies (Kidambi et al., 2018; Gitman et al., 2019; Yan et al., 2018).

The performance of SGD-based methods critically depends on hyperparameters such as the learning rate and batch size. Dynamic learning rate scheduling is widely adopted for improving training. For example, the cosine annealing schedule (Loshchilov & Hutter, 2017) smoothly reduces the learning rate, enhancing

both convergence speed and generalization. Additionally, increasing the batch size has been reported to improve the efficiency of mini-batch SGD (Shallue et al., 2019; Smith et al., 2018; Balles et al., 2017; Goyal et al., 2018; De et al., 2017). Recent theoretical studies, such as Umeda & Iiduka (2025) and Kamo & Iiduka (2025), further demonstrate that increasing the batch size can accelerate improvements in generalization performance and reductions in full gradient norms. Furthermore, Islamov et al. (2026) studied the role of batch size scaling in momentum-based conditional gradient methods, demonstrating that static batch sizes suffer from a critical saturation threshold, and they proposed an adaptive increasing schedule. While their setting differs from ours in both the algorithm (SCG/Scion) and the assumption ($\mu$-KL condition), the parallel finding that increasing batch sizes accelerates convergence provides a unified perspective on batch size scaling across different momentum methods.

While Umeda & Iiduka (2025) analyzed the convergence of vanilla SGD with scheduled learning rates and batch sizes without addressing momentum-based methods, Kamo & Iiduka (2025) studied the convergence of SGDM under a constant learning rate and increasing batch size but did not consider dynamic learning rate scheduling.

Although optimization-theoretic analyses of SGDM with dynamic learning rates remain limited, a complementary line of research models SGD as a diffusion process to study the interplay of dynamic learning rates and batch sizes on landscape exploration and exit times (Xie et al., 2021). Our work takes an optimization convergence perspective and complements this diffusion-theoretic approach by providing explicit convergence bounds under practical scheduling strategies. Sebbouh et al. (2021) studied the almost-sure convergence rates of SGD and SHB for smooth convex functions, providing an important theoretical foundation for momentum-based methods. The existing theoretical studies on SGDM primarily rely on constructing appropriate Lyapunov functions and leveraging their monotonic decrease to establish convergence (Gadat et al., 2018; Mai & Johansson, 2020; Wilson et al., 2021; Defazio, 2021). In particular, Liu et al. (2020) is a pioneering and influential work that, in the context of NSHB, derives a quantitative upper bound on the expected gradient norm of the objective function $f$ under a fixed learning rate, assuming only that $f$ is non-convex and $L$-smooth. However, extending such results to dynamic learning-rate schedules remains challenging.

In this work, we aim to fill this gap by analyzing the convergence of SGDM under dynamic learning rate scheduling.

**Our main contributions are summarized as follows:**

- We introduce a novel Lyapunov function for SGDM, enabling a rigorous convergence analysis that adapts to dynamic learning-rate schedules.

- We develop a unified theoretical framework covering both SHB and NSHB and derive convergence rates of the expected gradient norm under various scheduling strategies.

- We extend the analysis of Kamo & Iiduka (2025) from constant learning rates to decaying learning rates. Furthermore, we examine settings in which both the learning rate and batch size are increased and demonstrate that this leads to improved convergence. This setting has been studied by Umeda & Iiduka (2025) only for vanilla SGD.

- We validate our theoretical findings through experiments under four scheduling strategies and show that they align with theoretical predictions. The schedulings improve the full gradient norm in the following increasing order: (i) constant batch size with a decaying learning rate, (ii) increasing batch size with a decaying learning rate, (iii) increasing batch size with an increasing learning rate, and (iv) increasing batch size with a warmup learning rate.

## 2   Preliminaries

Let $\mathbb{N}$ denote the set of natural numbers. For any $n \in \mathbb{N}$, define $[n] := \{1, 2, \ldots, n\}$ and $[0 : n] := \{0, 1, \ldots, n\}$. Let $\mathbb{R}^d$ be a $d$-dimensional Euclidean space with inner product $\langle \boldsymbol{\theta}_1, \boldsymbol{\theta}_2 \rangle := \boldsymbol{\theta}_1^\top \boldsymbol{\theta}_2$ and norm $\|\boldsymbol{\theta}\| := \sqrt{\langle \boldsymbol{\theta}, \boldsymbol{\theta} \rangle}$.

Let $\mathbb{R}_+^d := \{\boldsymbol{\theta} \in \mathbb{R}^d \mid \theta_i \geq 0 \text{ for all } i \in [d]\}$ and $\mathbb{R}_{++}^d := \{\boldsymbol{\theta} \in \mathbb{R}^d \mid \theta_i > 0 \text{ for all } i \in [d]\}$. For scalars, define $\mathbb{R}_+ := \{x \in \mathbb{R} \mid x \geq 0\}$ and $\mathbb{R}_{++} := \{x \in \mathbb{R} \mid x > 0\}$.

Let $(x_t)$ and $(y_t)$ be sequences in $\mathbb{R}_+$. We write $y_t = O(x_t)$ if there exist constants $c \in \mathbb{R}_+$ and $t_0 \in \mathbb{N}$ such that $y_t \leq cx_t$ for all $t \geq t_0$.

## 2.1 Empirical Risk Minimization

Let $\boldsymbol{\theta} \in \mathbb{R}^d$ denote the parameters of a DNN. Let $S = \{(\boldsymbol{x}_1, \boldsymbol{y}_1), \ldots, (\boldsymbol{x}_n, \boldsymbol{y}_n)\}$ be a training dataset, where each $\boldsymbol{x}_i$ is associated with a label $\boldsymbol{y}_i$, and $n \in \mathbb{N}$ denotes the number of training samples. For each $(\boldsymbol{x}_i, \boldsymbol{y}_i)$, let the corresponding loss function be $f_i(\cdot) := f(\cdot; (\boldsymbol{x}_i, \boldsymbol{y}_i)) : \mathbb{R}^d \to \mathbb{R}$. The empirical loss is defined as $f(\boldsymbol{\theta}) := \frac{1}{n}\sum_{i \in [n]} f_i(\boldsymbol{\theta})$. Empirical risk minimization (ERM) aims to minimize this empirical loss. Note that this work focuses on finding a stationary point of the empirical loss, i.e., a point $\boldsymbol{\theta}^\star \in \mathbb{R}^d$ such that $\nabla f(\boldsymbol{\theta}^\star) = \mathbf{0}$.

We assume that the loss functions $f_i$ ($i \in [n]$) satisfy the conditions in the following assumption.

**Assumption 1.** *Let $n \in \mathbb{N}$ denote the number of training samples,*

*(A1) $f : \mathbb{R}^d \to \mathbb{R}$ is differentiable and $L$-smooth; i.e., there exists $L > 0$ such that, for all $\boldsymbol{\theta}_1, \boldsymbol{\theta}_2 \in \mathbb{R}^d$, $\|\nabla f(\boldsymbol{\theta}_1) - \nabla f(\boldsymbol{\theta}_2)\| \leq L\|\boldsymbol{\theta}_1 - \boldsymbol{\theta}_2\|$. The minimum value of $f$ is denoted by $f^\star := \inf_{\boldsymbol{\theta} \in \mathbb{R}^d} f(\boldsymbol{\theta}) > -\infty$.*

*(A2) Let $\xi$ be a random variable independent of $\boldsymbol{\theta}$. The stochastic gradient $\nabla f_\xi(\boldsymbol{\theta})$ satisfies:*

*(i) $\mathbb{E}_\xi[\nabla f_\xi(\boldsymbol{\theta})] = \nabla f(\boldsymbol{\theta})$ for all $\boldsymbol{\theta} \in \mathbb{R}^d$,*

*(ii) There exists $\sigma \geq 0$ such that for all $\boldsymbol{\theta} \in \mathbb{R}^d$, $\mathbb{V}_\xi[\nabla f_\xi(\boldsymbol{\theta})] = \mathbb{E}_\xi\left[\|\nabla f_\xi(\boldsymbol{\theta}) - \nabla f(\boldsymbol{\theta})\|^2\right] \leq \sigma^2$.*

*(A3) Let $b \in \mathbb{N}$ be a batch size with $b \leq n$, and let $\boldsymbol{\xi} = (\xi_1, \ldots, \xi_b)^\top$ be a vector of $b$ i.i.d. random variables, independent of $\boldsymbol{\theta}$. The mini-batch stochastic gradient is defined by $\nabla f_B(\boldsymbol{\theta}) := \frac{1}{b}\sum_{i=1}^b \nabla f_{\xi_i}(\boldsymbol{\theta})$, which is an unbiased estimator of the full gradient $\nabla f(\boldsymbol{\theta})$.*

## 2.2 Mini-batch SHB and NSHB Optimizers

Momentum-based stochastic methods are widely used to accelerate convergence and enhance stability. Here, we focus on two such methods:

- Mini-batch Stochastic Heavy Ball (SHB) (Polyak, 1964)

- Mini-batch Normalized SHB (NSHB) (Gupal & Bazhenov, 1972)

At each iteration $t$, given the current parameter $\boldsymbol{\theta}_t \in \mathbb{R}^d$, a mini-batch $\boldsymbol{\xi}_t = (\xi_{t,1}, \ldots, \xi_{t,b_t})^\top$ is sampled i.i.d. from $[n]$, and the mini-batch gradient is computed as $\nabla f_{B_t}(\boldsymbol{\theta}_t) := \frac{1}{b_t}\sum_{i=1}^{b_t} \nabla f_{\xi_{t,i}}(\boldsymbol{\theta}_t)$.

---

**Algorithm 1** Mini-batch NSHB

---

**Input**: Initial parameter $\boldsymbol{\theta}_0$
**Parameter**: Momentum coefficient $\beta \in [0, 1)$, learning rates $\{\eta_t\}_{t=0}^{T-1}$, batch sizes $\{b_t\}_{t=0}^{T-1}$, total steps $T$
**Output**: Final parameter $\boldsymbol{\theta}_T$

1: Initialize $\boldsymbol{m}_{-1} \leftarrow \mathbf{0}$
2: **for** $t = 0$ **to** $T - 1$ **do**
3: $\quad \nabla f_{B_t}(\boldsymbol{\theta}_t) \leftarrow \frac{1}{b_t}\sum_{i=1}^{b_t} \nabla f_{\xi_{t,i}}(\boldsymbol{\theta}_t)$
4: $\quad \boldsymbol{m}_t \leftarrow \beta \boldsymbol{m}_{t-1} + (1 - \beta)\nabla f_{B_t}(\boldsymbol{\theta}_t)$
5: $\quad \boldsymbol{\theta}_{t+1} \leftarrow \boldsymbol{\theta}_t - \eta_t \boldsymbol{m}_t$
6: **end for**
7: **return** $\boldsymbol{\theta}_T$

---

---

**Algorithm 2** Mini-batch SHB

---

**Input**: Initial parameter $\boldsymbol{\theta}_0$
**Parameter**: Momentum coefficient $\beta \in [0, 1)$, learning rates $\{\alpha_t\}_{t=0}^{T-1}$, batch sizes $\{b_t\}_{t=0}^{T-1}$, total steps $T$
**Output**: Final parameter $\boldsymbol{\theta}_T$

1: Initialize $\boldsymbol{m}_{-1} \leftarrow \boldsymbol{0}$
2: **for** $t = 0$ **to** $T - 1$ **do**
3:     $\nabla f_{B_t}(\boldsymbol{\theta}_t) \leftarrow \frac{1}{b_t} \sum_{i=1}^{b_t} \nabla f_{\xi_{t,i}}(\boldsymbol{\theta}_t)$
4:     $\boldsymbol{m}_t \leftarrow \beta \boldsymbol{m}_{t-1} + \nabla f_{B_t}(\boldsymbol{\theta}_t)$
5:     $\boldsymbol{\theta}_{t+1} \leftarrow \boldsymbol{\theta}_t - \alpha_t \boldsymbol{m}_t$
6: **end for**
7: **return** $\boldsymbol{\theta}_T$

---

Both algorithms can be equivalently rewritten in the following forms:

NSHB:

$$\boldsymbol{\theta}_{t+1} = \boldsymbol{\theta}_t - \eta_t(1-\beta)\nabla f_{B_t}(\boldsymbol{\theta}_t) + \beta \frac{\eta_t}{\eta_{t-1}}(\boldsymbol{\theta}_t - \boldsymbol{\theta}_{t-1}),$$

SHB:

$$\boldsymbol{\theta}_{t+1} = \boldsymbol{\theta}_t - \alpha_t \nabla f_{B_t}(\boldsymbol{\theta}_t) + \beta \frac{\alpha_t}{\alpha_{t-1}}(\boldsymbol{\theta}_t - \boldsymbol{\theta}_{t-1}).$$

Notably, NSHB reduces to SHB when $\eta_t = \frac{\alpha_t}{1-\beta}$.

## 3 Convergence Analysis of Mini-batch SGDM

As noted in the previous section, NSHB and SHB are equivalent up to the reparametrization $\eta_t = \frac{\alpha_t}{1-\beta}$. For notational consistency, we use $\lambda_t$ to denote the learning rate throughout this section. Therefore, we present the convergence analysis for NSHB only. The corresponding results for SHB follow analogously and are provided in Appendix A.

### 3.1 A Novel Lyapunov Function

In this section, we introduce the following Lyapunov function $\mathcal{L}_t$ to analyze the convergence of SGDM:

$$\mathcal{L}_t := \begin{cases} f(\boldsymbol{\theta}_t), & t = 0, \\ f(\boldsymbol{\theta}_t) + A_{t-1}\|\boldsymbol{m}_{t-1}\|^2, & t > 0, \end{cases} \tag{1}$$

where $A_t \geq 0$ is a deterministic scalar depending only on $t$. In particular, for the NSHB method, $A_t$ is defined as

$$A_t := \frac{\eta_t - L(1-\beta)\eta_t^2}{2(1-\beta)}.$$

A sketch explaining the appropriateness of this choice is provided later, and a detailed proof is deferred to Appendix C.2.

To contextualize our proposed Lyapunov function (1), we compare it with those of the prior studies. As summarized in Table 1, our formulation is significantly simpler. Moreover, it is highly versatile, as it can naturally accommodate dynamic learning-rate schedules.

### 3.2 General Convergence Bound

**Technical condition on learning rates:** To ensure the theoretical validity of the convergence analysis, we impose the following mild constraint on the variation of the learning rates:

$$\frac{\lambda_{t+1}}{\lambda_t} \leq c, \tag{2}$$

Table 1: Comparison of Lyapunov Functions for SGDM Convergence Analysis.
The key variables introduced in each work are defined as follows:
In (i), $\boldsymbol{z}_t := \frac{1}{1-\beta}\boldsymbol{\theta}_t - \frac{\beta}{1-\beta}\boldsymbol{\theta}_{t-1}$ is an auxiliary variable, and $\{c_i\}$ is a sequence of positive constants.
In (ii), $F_\mu$ is the Moreau envelope of $F = f + I_\mathcal{X}$, $\boldsymbol{p}_t := \frac{1-\beta}{\beta}(\boldsymbol{\theta}_t - \boldsymbol{\theta}_{t-1})$, $\boldsymbol{d}_t := \frac{1}{\eta}(\boldsymbol{\theta}_{t-1} - \boldsymbol{\theta}_t)$, and $\zeta := \frac{1-\beta}{\nu}$.
In (iii), $r_t$ controls the weight of past memory, and $a, b > 0$ ensure the Lyapunov function is non-negative and decreasing.
For further details, the reader is referred to the respective original works.
Note that Kamo & Iiduka (2025) uses the same form of Lyapunov function as (i).
(i): Liu et al. (2020), (ii): Mai & Johansson (2020), (iii): Gadat et al. (2018)

| Work | Lyapunov Function $\mathcal{L}_t$ | Function Properties | Learning Rate $\lambda_t$ |
|------|-----------------------------------|---------------------|---------------------------|
| **Ours** | $\mathcal{L}_t = f(\boldsymbol{\theta}_t) + A_{t-1}\|\boldsymbol{m}_{t-1}\|^2$ | $L$-smooth | Dynamic |
| (i) | $\mathcal{L}_t = f(\boldsymbol{z}_t) - f^\star + \sum_{i=1}^{t-1} c_i\|\boldsymbol{\theta}_{t+1-i} - \boldsymbol{\theta}_{t-i}\|^2$ | $L$-smooth | Constant |
| (ii) | $\mathcal{L}_t = F_\mu(\bar{\boldsymbol{\theta}}_t) + \frac{\nu\zeta^2}{4\mu^2}\|\boldsymbol{p}_t\|^2 + \frac{\lambda\zeta^2}{2\mu^2}\|\boldsymbol{d}_t\|^2 + \left(\frac{(1-\beta)\zeta^2}{2\mu^2} + \frac{\zeta}{\mu}\right)$ | weakly convex | Constant |
| (ii) | $\mathcal{L}_t = 2f(\boldsymbol{\theta}_t) + \frac{\varphi_t}{\nu\lambda^2} + \frac{\zeta}{2}\|\boldsymbol{d}_t\|^2$ | $L$-smooth | Constant |
| (iii) | $\mathcal{L}_t = (a + br_{t-1})f(\boldsymbol{\theta}_t) + \frac{a}{2r_{t-1}}\|\boldsymbol{m}_t\|^2 - b\langle\nabla f(\boldsymbol{\theta}_t), \boldsymbol{m}_t\rangle$ | $f \in C^2(\mathbb{R}^d)$, bounded Hessian, $\|\nabla f(\boldsymbol{\theta})\|^2 \leq cf(\boldsymbol{\theta})$ | $\lambda_t = \lambda/t^\beta$ |

for some constant $c$ satisfying $1 \leq c < \frac{1}{\beta^2}$. This condition accommodates both decaying and increasing learning-rate schedules:

- If the schedule is non-increasing, setting $c = 1$ implies $\lambda_{t+1} \leq \lambda_t$.

- If the schedule is non-decreasing, the learning rates may increase within the range satisfying $\lambda_t \leq \lambda_{t+1} \leq c\lambda_t$.

**Practical Limitations under Large Momentum Coefficients:** When the momentum coefficient $\beta$ is close to 1, which is standard in deep learning (e.g., $\beta = 0.9$ or $\beta = 0.99$), the condition (2) becomes highly restrictive. Specifically, $1/\beta^2 \approx 1.235$ for $\beta = 0.9$ and $1/\beta^2 \approx 1.020$ for $\beta = 0.99$, leaving very little room for the learning rate to increase. Furthermore, since $\lambda_{\max}$ is bounded by a term proportional to $1 - c\beta^2$ (see Theorem 1), a large momentum forces $\lambda_{\max}$ to be very small in order to maintain monotonic descent of our Lyapunov function. This presents a structural trade-off between momentum stability and aggressive learning rate scheduling.

The following theorem serves as the foundation for all subsequent theoretical results.

**Theorem 1** (General and unified convergence bound for NSHB). *Suppose Assumption 1 holds. Let $\{\boldsymbol{\theta}_t\}$ be the sequence generated by Algorithm 1 (NSHB) with learning rates $\lambda_t$ and batch sizes $b_t$. Assume*

$$\lambda_t \in [\lambda_{\min}, \lambda_{\max}] \subset \left[0, \frac{1 - c\beta^2}{L(1-\beta)}\right)$$

*and $\sum_{t=0}^{T-1} \lambda_t \neq 0$. Define*

$$B_T := \frac{1}{\sum_{t=0}^{T-1} \lambda_t}, \quad V_T := \frac{1}{\sum_{t=0}^{T-1} \lambda_t} \sum_{t=0}^{T-1} \frac{\lambda_t}{b_t}.$$

*Then, for any $T \in \mathbb{N}$, the following bound holds:*

$$\min_{0 \leq t \leq T-1} \mathbb{E}\left[\|\nabla f(\boldsymbol{\theta}_t)\|^2\right] \leq \frac{2(f(\boldsymbol{\theta}_0) - f^\star)}{1-\beta} B_T + \sigma^2 V_T,$$

*where $\mathbb{E}$ denotes the expectation over all randomness up to iteration $T$.*

**Remark 1** (Translation from squared gradient bound to gradient norm)**.** *The convergence bound in Theorem 1 controls the squared gradient norm $\mathbb{E}[\|\nabla f(\boldsymbol{\theta}_t)\|^2]$. If one wishes to report the bound in terms of the gradient norm $\mathbb{E}[\|\nabla f(\boldsymbol{\theta}_t)\|]$, it can be obtained directly via Jensen's inequality:*

$$\min_{0 \leq t \leq T-1} \mathbb{E}[\|\nabla f(\boldsymbol{\theta}_t)\|] \leq \sqrt{\min_{0 \leq t \leq T-1} \mathbb{E}[\|\nabla f(\boldsymbol{\theta}_t)\|^2]}.$$

*This remark clarifies that any statements or plots referring to the gradient norm (rather than squared norm) are consistent with Theorem 1.*

**Proof Sketch of Theorem 1** (The full proof is provided in Appendix C.2.)

To simplify the exposition, we focus on the case of NSHB with $\lambda_t = \eta_t$.

The main technical challenge in our analysis arises from the cross term,

$$\mathbb{E}[\langle \nabla f(\boldsymbol{\theta}_t), \boldsymbol{m}_{t-1} \rangle],$$

which appears when applying the $L$-smoothness of $f$ to the upper bound $f(\boldsymbol{\theta}_{t+1})$, but it is difficult to evaluate directly.

To resolve this issue, we introduce the Lyapunov function (for simplicity, in this sketch we consider only the case $t > 0$ in (1)),

$$\mathcal{L}_t := f(\boldsymbol{\theta}_t) + A_{t-1}\|\boldsymbol{m}_{t-1}\|^2,$$

and evaluate its expected difference:

$$\mathbb{E}[\mathcal{L}_{t+1} - \mathcal{L}_t] = \mathbb{E}[f(\boldsymbol{\theta}_{t+1}) - f(\boldsymbol{\theta}_t)] + A_t\mathbb{E}[\|\boldsymbol{m}_t\|^2] - A_{t-1}\mathbb{E}[\|\boldsymbol{m}_{t-1}\|^2].$$

To cancel out the cross terms appearing in the upper bounds of both $\mathbb{E}[f(\boldsymbol{\theta}_{t+1}) - f(\boldsymbol{\theta}_t)]$ and $A_t\mathbb{E}[\|\boldsymbol{m}_t\|^2]$, we define the coefficient $A_t$ as

$$A_t := \frac{\eta_t - L(1-\beta)\eta_t^2}{2(1-\beta)}.$$

This definition yields a tractable upper bound on $\mathbb{E}[\mathcal{L}_{t+1} - \mathcal{L}_t]$.

Excluding the influence of stochastic noise, the expected Lyapunov function decreases monotonically. Summing over $t$ and rearranging terms leads to the convergence bound stated in Theorem 1.

### 3.3 Constant Batch-size and Decaying Learning-rate Scheduler

First, we will consider the setting where the batch size remains fixed throughout training, while the learning rate follows a non-increasing schedule:

$$b_t = b, \quad \lambda_{t+1} \leq \lambda_t \quad (t \in \mathbb{N}). \tag{3}$$

Let $p > 0$ and $T, E \in \mathbb{N}$, with $0 \leq \lambda_{\min} \leq \lambda_{\max}$. Commonly used decaying learning-rate schedules include:

**[Constant LR]**

$$\lambda_t = \lambda_{\max}, \tag{4}$$

**[Diminishing LR]**

$$\lambda_t = \frac{\lambda_{\max}}{\sqrt{t+1}}, \tag{5}$$

**[Cosine LR]**

$$\lambda_t = \lambda_{\min} + \frac{\lambda_{\max} - \lambda_{\min}}{2}\left(1 + \cos\left(\left\lfloor \frac{t}{K} \right\rfloor \frac{\pi}{E}\right)\right), \tag{6}$$

**[Polynomial Decay LR]**

$$\lambda_t = (\lambda_{\max} - \lambda_{\min})\left(1 - \frac{t}{T}\right)^p + \lambda_{\min}, \tag{7}$$

where $K = \lceil n/b \rceil$ is the number of iterations per epoch, $E$ is the total number of epochs, and $T = KE$ in the cosine annealing schedule.

Applying Theorem 1 to the case of a constant batch size and the learning-rate schedules in (3), we obtain the following explicit convergence bounds. The proof is provided in Appendix C.3.

**Corollary 1** (Convergence rates under schedule (3)). *Under the assumptions of Theorem 1, suppose Algorithm 1 (NSHB) is run with a constant batch size $b_t \equiv b$ and a learning-rate schedule $\{\lambda_t\}$ satisfying (3). Then, the quantities $B_T$ and $V_T$ defined in Theorem 1 satisfy*

$$B_T \leq \begin{cases} \dfrac{1}{\lambda_{\max}T}, & [\textit{Constant LR (4)}] \\[2ex] \dfrac{1}{2\lambda_{\max}(\sqrt{T+1}-1)}, & [\textit{Diminishing LR (5)}] \\[2ex] \dfrac{2}{(\lambda_{\min}+\lambda_{\max})T}, & [\textit{Cosine LR (6)}] \\[2ex] \dfrac{p+1}{(p\lambda_{\min}+\lambda_{\max})T}, & [\textit{Polynomial LR (7)}] \end{cases} \qquad V_T = \frac{1}{b}.$$

*As a result, the expected gradient norm under NSHB satisfies*

$$\min_{0\leq t\leq T-1} \mathbb{E}[\|\nabla f(\boldsymbol{\theta}_t)\|] = \begin{cases} O\left(\sqrt{\dfrac{1}{T} + \dfrac{1}{b}}\right) & \begin{bmatrix}\textit{Constant LR (4)} \\ \textit{Cosine LR (6)} \\ \textit{Polynomial LR (7)}\end{bmatrix}, \\[3ex] O\left(\sqrt{\dfrac{1}{\sqrt{T}} + \dfrac{1}{b}}\right) & [\textit{Diminishing LR (5)}]. \end{cases}$$

**Interpretation of the Variance Floor and Technical Limitations:** From Corollary 1, the variance term $V_T = 1/b$ remains as a constant floor that does not vanish as $T \to \infty$. We emphasize that this is a limitation of our specific Lyapunov analysis, not an inherent property of SGDM.

In conventional analyses, the variance bound scales with the squared step size (e.g., $O(\eta^2/b)$ under a constant learning rate (Liu et al., 2020; Kamo & Iiduka, 2025), or $\sum \lambda_t^2/b_t$ under dynamic schedules (Umeda & Iiduka, 2025)), which vanishes as the step size decreases. In contrast, our Lyapunov function (1) cancels the cross-term $\mathbb{E}[\langle \nabla f(\boldsymbol{\theta}_t), \boldsymbol{m}_{t-1} \rangle]$ via the coefficient $A_t$, at the cost of a variance term $V_T$ that depends linearly on $\lambda_t$ rather than $\lambda_t^2$. When $b_t \equiv b$, the learning rates cancel out, leaving $V_T = 1/b$ regardless of the decay schedule.

Therefore, our framework requires a dynamically increasing batch size to achieve convergence to zero; this represent a trade-off between proof simplicity and strict convergence under constant batch sizes.

### 3.4 Increasing Batch-size and Decaying Learning-rate Scheduler

Next, we consider the setting where the batch size increases over time while the learning rate decreases, i.e.,

$$b_t \leq b_{t+1}, \quad \lambda_{t+1} \leq \lambda_t \quad (t \in \mathbb{N}). \tag{8}$$

We introduce the total number of phases $M \in \mathbb{N}$ (also referred to as the number of batch size updates) up front and index the phases by $m = 0, \ldots, M-1$. For each phase $m$, let $E_m \in \mathbb{N}$ denote the number of epochs in phase $m$, and let $K_m \in \mathbb{N}$ denote the number of steps per epoch in that phase. Therefore, the $m$-th phase contains $K_m E_m$ iterations. For convenience, we define the phase-indexed step sets

$$S_m := \left\{ t \in \mathbb{N} \,\middle|\, \sum_{k=0}^{m-1} K_k E_k \leq t < \sum_{k=0}^{m} K_k E_k \right\},$$

with the convention $\sum_{k=0}^{-1}(\cdot) = 0$. The total number of iterations is

$$T := \sum_{m=0}^{M-1} K_m E_m. \tag{9}$$

For any $m$ and $t \in S_m$, the batch size $b_t$ may be specified in a phase-wise manner.

**[Exponentially Growing Batch Size]**

$$b_t = \delta^{m \left\lceil \frac{t}{\sum_{k=0}^{m} K_k E_k} \right\rceil} b_0, \tag{10}$$

where $\delta > 1$ is the growth factor and $b_0 > 0$ is the initial batch size. For example, setting $\delta = 2$ corresponds to doubling the batch size at each phase. Specifically, the sequence of batch sizes can be represented as

$$\underbrace{b_0 \delta^0, \ldots, b_0 \delta^0}_{K_0 E_0}, \underbrace{b_0 \delta^1, \ldots, b_0 \delta^1}_{K_1 E_1}, \ldots, \underbrace{b_0 \delta^m, \ldots, b_0 \delta^m}_{K_m E_m}, \ldots, \underbrace{b_0 \delta^{M-1}, \ldots, b_0 \delta^{M-1}}_{K_{M-1} E_{M-1}}.$$
$$\underbrace{\phantom{b_0 \delta^0, \ldots, b_0 \delta^0, b_0 \delta^1, \ldots, b_0 \delta^1, \ldots, b_0 \delta^m, \ldots, b_0 \delta^m, \ldots, b_0 \delta^{M-1}, \ldots, b_0 \delta^{M-1}}}_{T = \sum_{m=0}^{M-1} K_m E_m}$$

This phase-based schedule follows Smith et al. (2018); Umeda & Iiduka (2025); Kamo & Iiduka (2025).

Applying Theorem 1 to the setting of increasing batch sizes and decaying learning rates in (8), we derive the following convergence bounds. The proof is given in Appendix C.4.

**Corollary 2** (Convergence rates under schedule (8))**.** *Under the assumptions of Theorem 1, suppose Algorithm 1 (NSHB) is run with batch sizes $\{b_t\}$ and learning rates $\{\lambda_t\}$ following (8). For $M \in \mathbb{N}$, let $T = \sum_{m=0}^{M-1} K_m E_m$, $E_{\max} = \sup_m E_m$, and $K_{\max} = \sup_m K_m$. Then, $B_T$ and $V_T$ from Theorem 1 satisfy the bounds below, where the bound for $B_T$ is defined in Corollary 1, and $V_T$ is bounded by:*

$$V_T \leq \begin{cases} \dfrac{\delta K_{\max} E_{\max}}{(\delta-1) b_0 T}, & [\text{Constant LR (4)}], \\[2mm] \dfrac{\delta K_{\max} E_{\max}}{2(\delta-1) b_0 (\sqrt{T+1}-1)}, & [\text{Diminishing LR (5)}], \\[2mm] \dfrac{2\delta \lambda_{\max} K_{\max} E_{\max}}{(\delta-1)(\lambda_{\min}+\lambda_{\max}) b_0 T}, & [\text{Cosine LR (6)}], \\[2mm] \dfrac{(p+1)\delta \lambda_{\max} K_{\max} E_{\max}}{(\delta-1)(\lambda_{\max}+p\lambda_{\min}) b_0 T}, & [\text{Polynomial LR (7)}]. \end{cases}$$

*As a result, the expected gradient norm under NSHB satisfies*

$$\min_{0 \leq t \leq T-1} \mathbb{E}[\|\nabla f(\boldsymbol{\theta}_t)\|] = \begin{cases} O\left(\dfrac{1}{\sqrt{T}}\right), & \left[\begin{array}{c}\text{Constant LR (4), Cosine LR (6),} \\ \text{Polynomial LR (7)}\end{array}\right], \\[3mm] O\left(\dfrac{1}{T^{1/4}}\right), & [\text{Diminishing LR (5)}]. \end{cases}$$

Since the batch size $b_t$ increases over time, the variance term $V_T = \frac{1}{\sum_{t=0}^{T-1} \lambda_t} \sum_{t=0}^{T-1} \frac{\lambda_t}{b_t}$ vanishes as $T \to \infty$, unlike in the constant batch size case. Consequently, we have

$$\min_{0 \leq t \leq T-1} \mathbb{E}\left[\|\nabla f(\boldsymbol{\theta}_t)\|\right] \to 0 \quad \text{as } T \to \infty,$$

showing that using growing batch sizes during training can remove the variance floor and guarantee convergence under suitable learning rates.

Next, we consider the case where the batch size is updated $M$ times. In this case, the total number of iterations is given by (9) as $T = \sum_{m=0}^{M-1} K_m E_m$. Here, the number of iterations per phase satisfies $K_m \geq 1$, and we define $E_{\min} := \inf_m E_m > 0$. Then, the following inequality holds:

$$T = \sum_{m=0}^{M-1} K_m E_m \geq \sum_{m=0}^{M-1} E_m \geq M E_{\min}.$$

Therefore, the convergence rate with respect to the number of updates $M$ is

$$\min_{0 \leq t \leq T-1} \mathbb{E}[\|\nabla f(\boldsymbol{\theta}_t)\|] = O\left(\frac{1}{\sqrt{T}}\right) = O\left(\frac{1}{\sqrt{M}}\right). \tag{11}$$

### 3.5 Increasing Batch-size and Increasing Learning-rate Scheduler

Next, we consider the setting in which both the batch size and the learning rate increase over time:

$$b_t \leq b_{t+1}, \quad \lambda_t \leq \lambda_{t+1} \quad (t \in \mathbb{N}). \tag{12}$$

For any $m$ and $t \in S_m$, the batch sizes and learning rates are, for example, given by:

[**Exponential Growth of Batch Size and Learning Rate**]

$$b_t = \delta^{m\left\lceil \frac{t}{\sum_{k=0}^{m} K_k E_k} \right\rceil} b_0, \quad \lambda_t = \gamma^{m\left\lceil \frac{t}{\sum_{k=0}^{m} K_k E_k} \right\rceil} \lambda_0, \tag{13}$$

where $\delta, \gamma > 1$ with $\gamma < \delta$. Here, $b_0 > 0$ and $\lambda_0 > 0$ denote the initial batch size and learning rate, respectively.

Applying Theorem 1 to the setting defined by (12) and the learning rate growth condition (2), we obtain the following convergence bounds. The proof is given in Appendix C.5.

**Corollary 3** (Convergence rates under schedule (12)). *Under the assumptions of Theorem 1, suppose Algorithm 1 (NSHB) is run with batch sizes $\{b_t\}$ and learning rates $\{\lambda_t\}$ satisfying (12) and (2). For all $M \in \mathbb{N}$, here, $T$, $E_{\max}$, and $K_{\max}$ are as defined in Corollary 2. Let $E_{\min} = \inf_{M \in \mathbb{N}} \inf_{m \in [0:M-1]} E_m < +\infty$, $K_{\min} = \inf_{M \in \mathbb{N}} \inf_{m \in [0:M-1]} K_m < +\infty$, and $\hat{\gamma} = \frac{\gamma}{\delta} < 1$. Then, the quantities $B_T$ and $V_T$ in Theorem 1 satisfy the bounds:*

$$B_T \leq \frac{\delta^2}{\lambda_0 K_{\min} E_{\min} \gamma^M}, \qquad V_T \leq \frac{K_{\max} E_{\max} \lambda_0 \delta^2}{K_{\min} E_{\min} b_0 (1 - \hat{\gamma}) \gamma^M}.$$

*As a result, the expected gradient norm under NSHB satisfies*

$$\min_{0 \leq t \leq T-1} \mathbb{E}[\|\nabla f(\boldsymbol{\theta}_t)\|] = O\left(\frac{1}{\gamma^{M/2}}\right).$$

This result shows that using exponentially increasing batch sizes and learning rates yields an exponentially fast decay in the expected gradient norm.

In contrast, under the schedule with increasing batch sizes and a decaying learning rate in (8), the convergence rate remains polynomial, as shown in (11), and scales as $O(1/\sqrt{M})$ with respect to the number of updates $M$. The exponential schedule (12), however, achieves a much faster convergence rate of $O(\gamma^{-M/2})$, which is asymptotically superior.

## 4 Experiments

We evaluated two momentum-based optimization algorithms—stochastic heavy ball (SHB) and its normalized variant (NSHB)—on CIFAR-100 using ResNet-18. Unless stated otherwise, we set the momentum

coefficient to $\beta = 0.9$ and trained all models for 300 epochs. The experiments were conducted on a system equipped with dual Intel Xeon Silver 4316 CPUs and NVIDIA Tesla A100 80GB GPUs. The software environment consisted of Python 3.8.2, CUDA 12.2, and PyTorch 2.4.1.

We considered the following four training schedules:

   (i) Constant batch size with decaying learning rate,

  (ii) Increasing batch size with decaying learning rate,

 (iii) Increasing batch size with increasing learning rate,

 (iv) Increasing batch size with warmup learning rate.

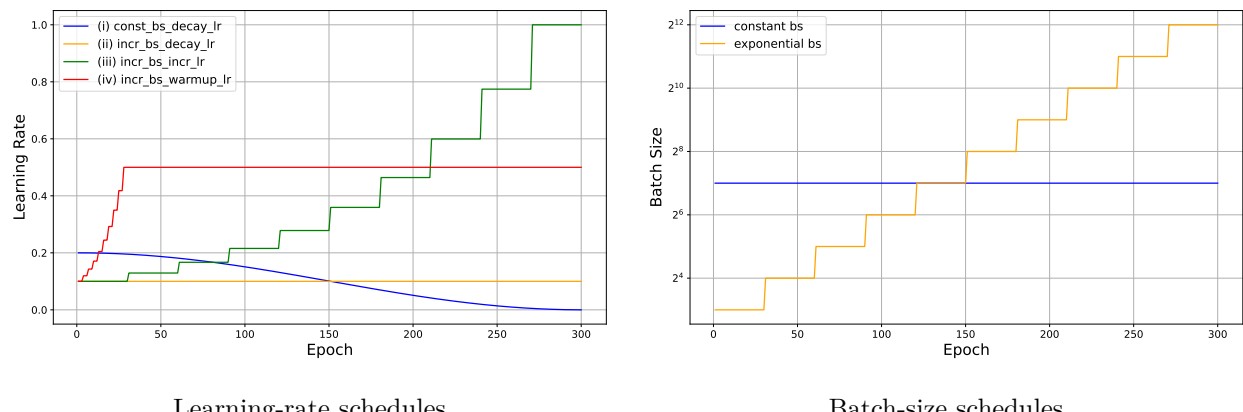

Learning-rate schedules                             Batch-size schedules

Figure 1: Learning rate (left) and batch size (right) schedules for the four training configurations (i)–(iv) used in our experiments. Results for other learning rate schedules (e.g., diminishing (5) and polynomial decay (7)) are provided in Appendix D.

**On the Synchronization Mismatch between Theory and Experiments:** Our theoretical framework assumes that the learning rate $\lambda_t$ and batch size $b_t$ are updated synchronously at phase boundaries. In the warmup schedule (iv), however, these updates are decoupled: the learning rate is updated at 3-epoch intervals during the warmup phase, while the batch size scales up at 30-epoch intervals. This choice is motivated by the critical batch size (Shallue et al., 2019; Sato & Iiduka, 2023; Imaizumi & Iiduka, 2025a): increasing the batch size as frequently as the learning rate risks exceeding the critical threshold prematurely, causing most of the training to be conducted at a very large batch size, which can degrade performance. The synchronized update setting is analyzed in Appendix B.

The solid lines in the figures show the mean over three runs; the shaded areas indicate the range between the maximum and minimum values. We report the training loss, test accuracy, and full gradient norm $\|\nabla f(\boldsymbol{\theta}_e)\|$ versus the number of epochs, where $\boldsymbol{\theta}_e$ denotes the model parameters at the end of each epoch. The full gradient norm was computed at the end of each epoch by averaging the gradients over all mini-batches to reconstruct the gradient over the entire training set and subsequently taking its L2 norm.

Figures 2–3 summarize representative training dynamics under the four schedules considered in this work; detailed settings and additional runs are provided in Appendix D. With respect to the gradient norm, both methods show the same ordering across schedules: (i) worst, (ii) intermediate, (iii) better intermediate, and (iv) best; this corroborates our theoretical predictions. As for test accuracy, largely orthogonal to convergence guarantees, both methods show similar trends: increasing the batch size generally improves generalization, and schedule (iv) attains the best accuracy. Additionally, for SHB, a trade-off is observed between learning rate regimes: the high learning rate reduces the gradient norm faster, whereas the low learning rate yields better test accuracy.

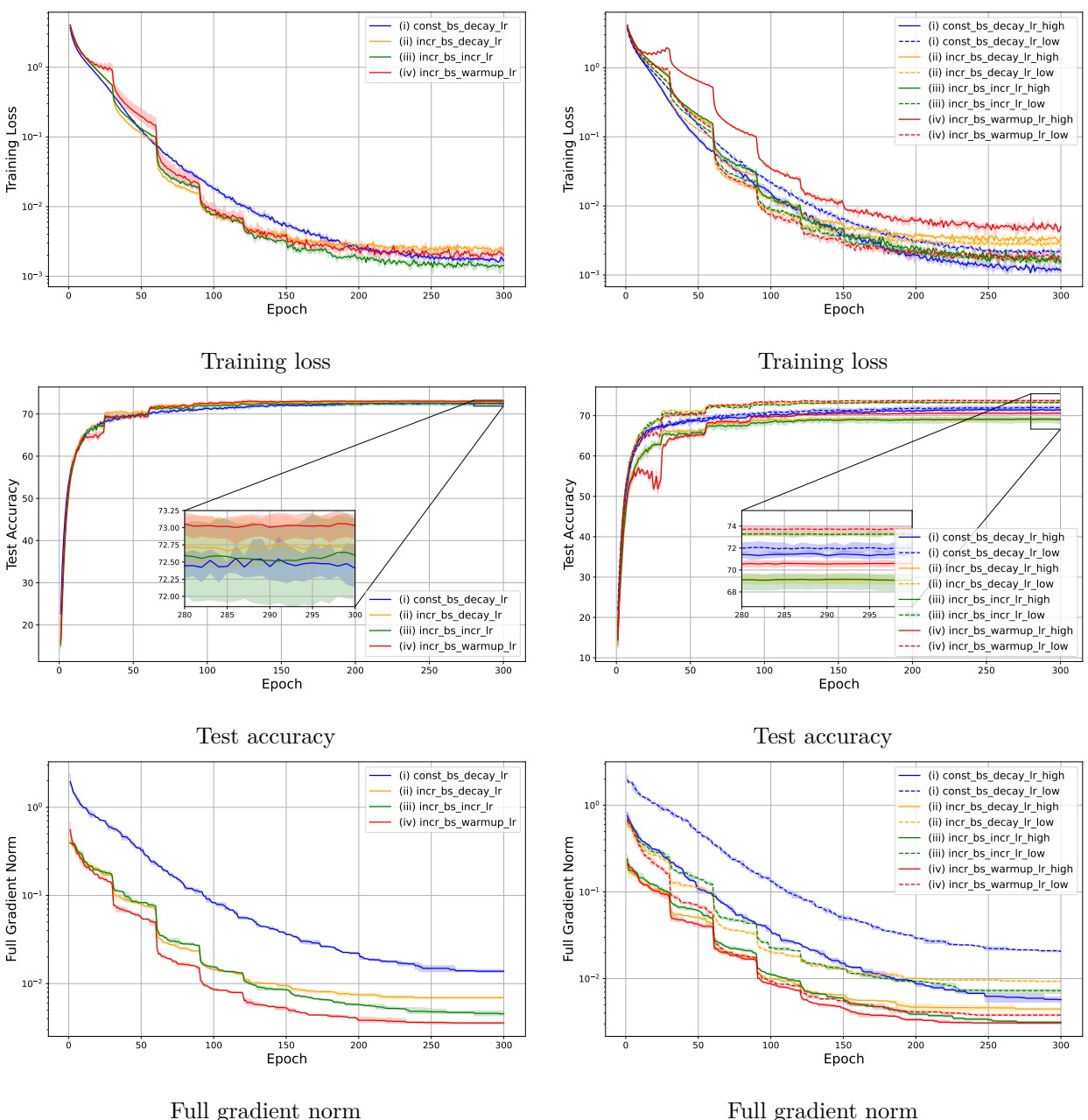

Figure 2: Results of NSHB: representative training dynamics. Each panel shows (top) training loss, (middle) test accuracy, and (bottom) gradient norm. For clarity, one representative instance is shown for each of four schedules; schedule (i) uses the Cosine LR (6), and schedule (ii) uses the Constant LR (4). Full results for all schedules and additional runs are provided in Appendix D.1.

Figure 3: Results of SHB: representative training dynamics. Solid lines indicate the high learning rate setting (same LR as NSHB), and dashed lines indicate the low learning rate setting (one-tenth of the NSHB LR), in consideration of the theoretical constraints in Theorem 1. See Appendix D.2 for full results and additional settings.

Figure 4 compares NSHB and SHB with commonly used optimizers (SGD, RMSProp, Adam, and AdamW) under the increasing batch-size schedule from (ii). In terms of optimization dynamics, SGD, NSHB, and SHB exhibit a rapid decrease in gradient norms during the early stages. However, in the later stages, Adam

achieves smaller gradient norms. This tendency has also been reported by Kamo & Iiduka (2025), and we anticipate further theoretical analyses of Adam under increased batch sizes.

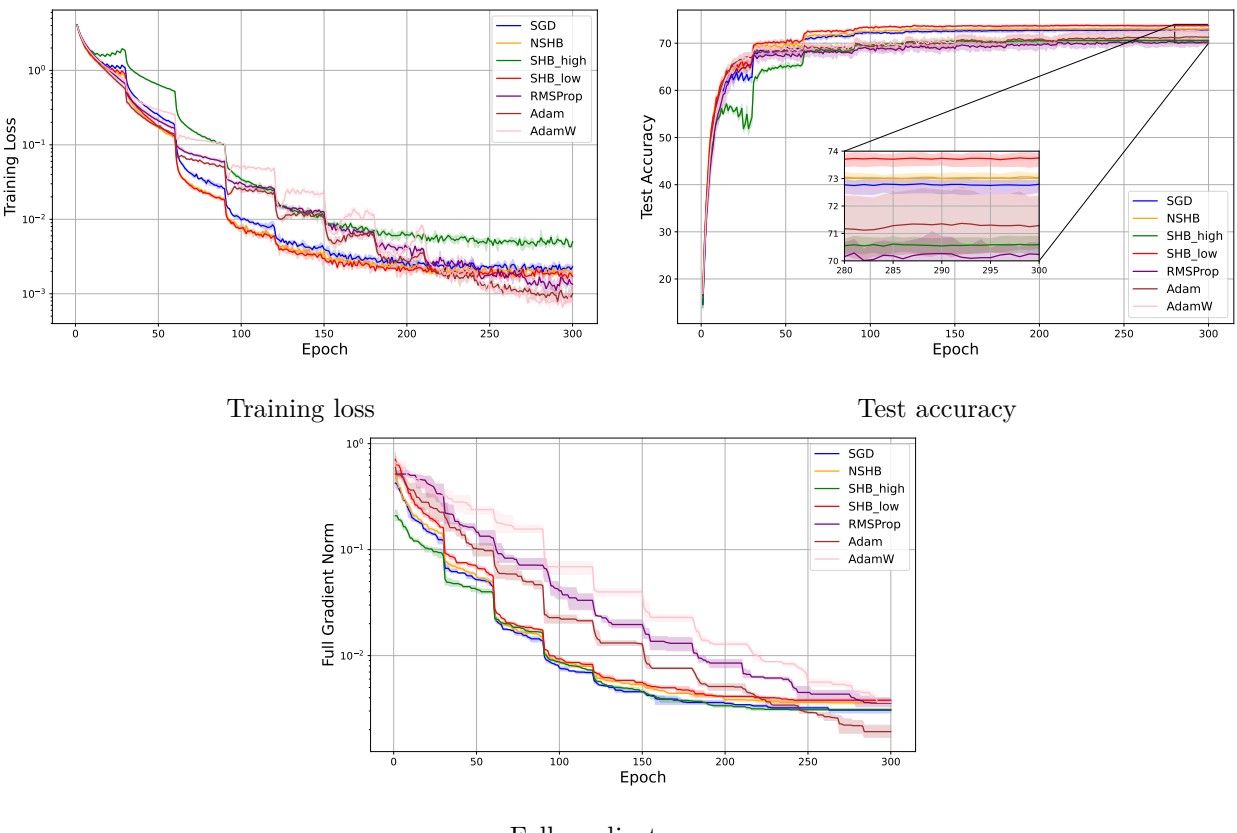

Training loss                                                              Test accuracy

Full gradient norm

Figure 4: Comparison of NSHB and SHB with common optimizers under the increasing batch-size schedule from (ii). SGD uses the warmup schedule from (iv); RMSProp, Adam, and AdamW use LR=0.0005. Optimizer hyperparameters are as follows: RMSProp ($\beta_2 = 0.99$), Adam/AdamW ($\beta_1 = 0.9$, $\beta_2 = 0.999$); AdamW additionally uses a weight decay of 0.01.

# 5    Discussion

In this section, we discuss the technical challenges in extending existing SGDM analyses to dynamic schedules, interpret the empirical results from the perspective of escape from local minima, and present an extension to dynamic momentum scheduling.

## 5.1    Technical Challenges in Extending SGDM Analysis to Dynamic Schedules

Under a constant learning rate $\eta$, Liu et al. (2020) introduces the auxiliary variable $\boldsymbol{z}_t := \frac{1}{1-\beta}\boldsymbol{\theta}_t - \frac{\beta}{1-\beta}\boldsymbol{\theta}_{t-1}$, which gives $\boldsymbol{z}_{t+1} - \boldsymbol{z}_t = -\eta\nabla f_{B_t}(\boldsymbol{\theta}_t)$, eliminating the momentum term. Under a dynamic learning rate $\eta_t$, however, this becomes:

$$\boldsymbol{z}_{t+1} - \boldsymbol{z}_t = -\eta_t\nabla f_{B_t}(\boldsymbol{\theta}_t) - \frac{\beta}{1-\beta}(\eta_t - \eta_{t-1})\boldsymbol{m}_{t-1},$$

introducing a residual term that prevents direct application of this approach. Without such a variable transformation, the cross-term $\mathbb{E}[\langle\nabla f(\boldsymbol{\theta}_t), \boldsymbol{m}_{t-1}\rangle]$ remains in the descent bound and its sign is indefinite

across iterations, complicating standard analysis. Our framework resolves this by choosing the coefficient $A_t$ in $\mathcal{L}_t$ to cancel this cross-term at each iteration.

## 5.2 Escape from Sharp Minima via Increasing Batch Size

The dynamic scheduling of learning rates and batch sizes can be interpreted from the perspective of escape from local minima. Under the theoretical framework of Xie et al. (2021), the magnitude of stochastic noise scales with the ratio $\lambda_t/b_t$. Our exponential growth schedules satisfy $\lambda_t/b_t \to 0$ as $T \to \infty$, which induces a two-phase effect analogous to simulated annealing:

1. **Early Training:** A small batch size maintains a large $\lambda_t/b_t$ ratio, introducing strong noise that facilitates escape from sharp local minima with poor generalization.

2. **Late Training:** The exponentially increasing batch size drives $\lambda_t/b_t$ toward 0, suppressing noise fluctuations and allowing parameters to stably settle into flatter, more generalizable minima.

This interpretation is supported by the numerical evaluations of Imaizumi & Iiduka (2025b) for momentum methods with increasing batch sizes. Their results show that large constant batch sizes yield worse generalization than small constant batch sizes, while an exponentially increasing batch-size schedule outperforms both. This closely aligns with our interpretation, wherein a small initial batch size enables escape from sharp minima and the subsequent increase stabilizes convergence toward flat minima. Therefore, their findings provide strong empirical support for the practical success of schedules (ii), (iii), and (iv).

## 5.3 Extension to Dynamic Momentum Scheduling

It is well known that SGDM's performance strongly depends not only on the learning rate but also on the momentum coefficient $\beta$ (Shi, 2024). Several recent studies (Chen et al., 2022; Li et al., 2025) have demonstrated the effectiveness of dynamic momentum scheduling in practice. Our Lyapunov-based analysis naturally extends to the case where $\beta_t$ is monotonically non-increasing. Specifically, by redefining the Lyapunov coefficient as

$$A_t := \frac{\eta_t - L(1 - \beta_t)\eta_t^2}{2(1 - \beta_t)},$$

the same descent argument applies, and the convergence bound becomes

$$\min_{0 \le t \le T-1} \mathbb{E}\left[\|\nabla f(\boldsymbol{\theta}_t)\|^2\right] \le \frac{2(f(\boldsymbol{\theta}_0) - f^\star)}{\sum_{t=0}^{T-1}(1 - \beta_t)\eta_t} + \frac{\sigma^2}{\sum_{t=0}^{T-1}(1 - \beta_t)\eta_t} \sum_{t=0}^{T-1} \frac{(1 - \beta_t)\eta_t}{b_t}.$$

Note that this reduces to Theorem 1 when $\beta_t = \beta$ is fixed. The details are provided in Appendix E.

## 6 Conclusion

In this paper, we extended the existing theoretical analyses of the learning rate and batch size scheduling for mini-batch SGD to the SGDM framework. By introducing a newly constructed Lyapunov function, we provided a unified analysis of widely used momentum-based optimization training methods. Consequently, we established convergence guarantees on the expected full gradient norm of the empirical loss. Theoretically, we showed that an increasing batch-size schedule ensures convergence of SGDM. and that simultaneously increasing both the batch size and the learning rate can accelerate convergence.

We summarize our main contributions as follows:

- We introduced a novel and simpler Lyapunov function for SGDM that unifies the convergence analysis of SHB and NSHB under dynamic learning-rate and batch-size schedules.

- We extended the analysis of Kamo & Iiduka (2025) from constant to decaying learning rates, as well as extended the increasing learning-rate and batch-size setting of Umeda & Iiduka (2025), originally studied only for vanilla SGD, to the momentum setting.

- We validated our theoretical findings through experiments confirming that the full gradient norm improves progressively across four scheduling strategies, from a constant batch size with a decaying learning rate to an increasing batch size combined with a warmup learning rate.

**Acknowledgments**

We are sincerely grateful to the Action Editor, Antonio Orvieto, and the three anonymous reviewers for helping us improve the original manuscript. This research is partly supported by the computational resources of the DGX A100 named TAIHO at Meiji University. This work was supported by the Japan Society for the Promotion of Science (JSPS) KAKENHI Grant Number 24K14846 awarded to Hideaki Iiduka.

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

## A   General Convergence Bound for SHB

As stated in Section 3, the stochastic heavy ball (SHB) method is equivalent to the normalized version (NSHB) up to the reparametrization $\eta_t = \frac{\alpha_t}{1-\beta}$, where $\alpha_t$ and $\eta_t$ denote the learning rates of SHB and NSHB, respectively. For completeness, this section presents an explicit general convergence bound for SHB derived under this equivalence, corresponding to Theorem 1 in the main text.

**Theorem 2** (General and unified convergence bound for SHB)**.** *Suppose Assumption 1 holds. Let $\{\boldsymbol{\theta}_t\}$ be the sequence generated by Algorithm 2 (SHB) with learning rates $\lambda_t$ and batch sizes $b_t$. Assume*

$$\lambda_t \in [\lambda_{\min}, \lambda_{\max}] \subset \left[0, \frac{1 - c\beta^2}{L}\right)$$

*and $\sum_{t=0}^{T-1} \lambda_t \neq 0$. Define*

$$B_T := \frac{1}{\sum_{t=0}^{T-1} \lambda_t}, \quad V_T := \frac{1}{\sum_{t=0}^{T-1} \lambda_t} \sum_{t=0}^{T-1} \frac{\lambda_t}{b_t}.$$

*Then, for any $T \in \mathbb{N}$, the following bound holds:*

$$\min_{0 \leq t \leq T-1} \mathbb{E}\left[\|\nabla f(\boldsymbol{\theta}_t)\|^2\right] \leq 2(f(\boldsymbol{\theta}_0) - f^\star) B_T + \sigma^2 V_T,$$

*where $\mathbb{E}$ denotes the expectation over all randomness up to iteration $T$.*

Since the mathematical structure of the convergence bound is identical to Theorem 1 except for the constants scaled by the reparametrization, all subsequent schedule-specific results (Corollaries 1–4) follow analogously for SHB.

## B   Increasing Batch Size with Warmup Learning Rate Scheduler

In the main experiments, the batch size was increased every 30 epochs, while the learning rate followed a more frequent warmup schedule. This setting does not satisfy the scheduler defined in equation (13), which updates both the batch size and the learning rate in the same epoch and forms the basis of our theoretical analysis. Here, we provide an extended theoretical analysis under the stricter assumption of synchronized updates to ensure clarity and completeness. Although the experiments deviate from these assumptions, the theoretical results offer valuable insight.

Let the warmup period last for $T_w = \sum_{m=0}^{M_w} K_m E_m > 0$ iterations, corresponding to $M_w$ phases of increasing learning rate. The schedule satisfies:

$$\begin{aligned} b_t &\leq b_{t+1} \quad (t \in \mathbb{N}), \\ \lambda_t &\leq \lambda_{t+1} \quad (t < T_w), \quad \lambda_{t+1} \leq \lambda_t \quad (t \geq T_w). \end{aligned} \tag{14}$$

The increase in the learning rate during the warmup period must satisfy the growth condition in (2).

The batch sizes $\{b_t\}$ follow the exponential growth schedule in (10), and the learning rates $\{\lambda_t\}$ follow warmup variants of the constant and cosine decay schedules, for all $m \in [0 : M]$ and all $t \in S_m$, as follows:

**[Constant LR with Warmup]**

$$\lambda_t = \begin{cases} \gamma^{\left\lceil \frac{t}{\sum_{k=0}^{m} K_k E_k} \right\rceil} \lambda_0 & (m \in [0 : M_w]), \\ \gamma^{M_w} \lambda_0 & (m \in [M_w : M]), \end{cases} \tag{15}$$

**[Cosine LR with Warmup]**

$$
\lambda_t = \begin{cases}
\gamma^{m\left\lceil \frac{t}{\sum_{k=0}^{m} K_k E_k} \right\rceil} \lambda_0 & (m \in [0 : M_w]), \\
\lambda_{\min} + \dfrac{\lambda_{\max} - \lambda_{\min}}{2} \\
\times \left( 1 + \cos\left( \left[ \sum_{k=0}^{m-1} E_k + \left\lfloor \dfrac{t - \sum_{k=0}^{m-1} K_k E_k}{K_m} \right\rfloor - E_w \right] \dfrac{\pi}{E_M - E_w} \right) \right) & (m \in [M_w : M]),
\end{cases}
\tag{16}
$$

where $E_w$ is the number of warmup epochs, $\lambda_{\max} := \gamma^{M_w} \lambda_0$, and $\gamma > 1$ is the learning rate growth factor during warmup.

By applying Theorem 1 to the combined schedule in (14) with exponentially increasing batch sizes (10), we obtain the following convergence bounds. The proof is given in Appendix C.6.

**Corollary 4** (Convergence rates under schedule (14)). *Under the assumptions of Theorem 1, suppose Algorithm 1 (NSHB) or Algorithm 2 (SHB) is run with batch sizes $\{b_t\}$ and learning rates $\{\lambda_t\}$ defined by the warmup schedule (14). Let $\delta, \gamma > 1$ with $\hat{\gamma} = \frac{\gamma}{\delta} < 1$. Here, $T$, $E_{\max}$, $E_{\min}$, $K_{\max}$, and $K_{\min}$ are as defined in Corollaries 2–3. Then, the quantities $B_T$ and $V_T$ in Theorem 1 satisfy the following bounds:*

$$
\text{[Constant LR (15)]} \quad
\begin{aligned}
B_T &\leq \frac{\delta^2}{\lambda_0 K_{\min} E_{\min} \gamma^{M_w}} + \frac{1}{\lambda_{\max}(T - T_w)}, \\
V_T &\leq \frac{K_{\max} E_{\max} \lambda_0 \delta^2}{K_{\min} E_{\min} b_0 (1 - \hat{\gamma}) \gamma^{M_w}} + \frac{\delta K_{\max} E_{\max}}{(\delta - 1) b_0 (T - T_w)}.
\end{aligned}
$$

$$
\text{[Cosine LR (16)]} \quad
\begin{aligned}
B_T &\leq \frac{\delta^2}{\lambda_0 K_{\min} E_{\min} \gamma^{M_w}} + \frac{2}{(\lambda_{\min} + \lambda_{\max})(T - T_w)}, \\
V_T &\leq \frac{K_{\max} E_{\max} \lambda_0 \delta^2}{K_{\min} E_{\min} b_0 (1 - \hat{\gamma}) \gamma^{M_w}} + \frac{2\delta \lambda_{\max} K_{\max} E_{\max}}{(\delta - 1)(\lambda_{\min} + \lambda_{\max}) b_0 (T - T_w)}.
\end{aligned}
$$

*As a result, the expected gradient norm under both NSHB and SHB satisfies*

$$
\min_{t \in [T_w, T-1]} \mathbb{E}[\|\nabla f(\boldsymbol{\theta}_t)\|] = O\left( \frac{1}{\sqrt{T - T_w}} \right).
$$

In the phase $t \geq T_w$, both algorithms use increasing batch sizes and decaying learning rates, yielding convergence rates comparable to those under the decaying schedule (8) (see Corollary 2).

Importantly, the warmup phase for $t < T_w$ accelerates early-stage convergence by preventing premature decay of the learning rate. Thus, combining increasing batch sizes with warmup and decaying learning-rate schedules allows Algorithms 1 and 2 to reduce early-stage bias and later-stage variance, achieving faster overall convergence compared with decaying schedules alone.

# C  Proof

### C.1  Proposition

The following proposition summarizes the properties of the mini-batch gradient under the assumption of sampling with replacement, which is standard in the analysis of stochastic optimization (see, e.g., Bottou et al., 2018).

**Proposition 1.** *Let $t \in \mathbb{N}$, $\boldsymbol{\xi}_t$ be a random variable independent of $\boldsymbol{\xi}_j$ ($j \in [0 : t - 1]$), $\boldsymbol{\theta}_t \in \mathbb{R}^d$ be independent of $\boldsymbol{\xi}_t$, and $\nabla f_{B_t}(\boldsymbol{\theta}_t)$ be the mini-batch gradient, where $\nabla f_{\xi_{t,i}}$ ($i \in [b_t]$) is the stochastic gradient (see Assumption 1(A2)). Then, the following hold:*

$$
\mathbb{E}_{\boldsymbol{\xi}_t}\left[ \nabla f_{B_t}(\boldsymbol{\theta}_t) \Big| \hat{\boldsymbol{\xi}}_{t-1} \right] = \nabla f(\boldsymbol{\theta}_t), \quad \mathbb{V}_{\boldsymbol{\xi}_t}\left[ \nabla f_{B_t}(\boldsymbol{\theta}_t) \Big| \hat{\boldsymbol{\xi}}_{t-1} \right] \leq \frac{\sigma^2}{b_t},
$$

where $\mathbb{E}_{\boldsymbol{\xi}_t}[\cdot|\hat{\boldsymbol{\xi}}_{t-1}]$ and $\mathbb{V}_{\boldsymbol{\xi}_t}[\cdot|\hat{\boldsymbol{\xi}}_{t-1}]$ are respectively the expectation and variance with respect to $\boldsymbol{\xi}_t$ conditioned on $\boldsymbol{\xi}_{t-1} = \hat{\boldsymbol{\xi}}_{t-1}$.

*Proof.* Assumption 1(A3) and the independence of $b_t$ and $\boldsymbol{\xi}_t$ ensure that

$$\mathbb{E}_{\boldsymbol{\xi}_t}\left[\nabla f_{B_t}(\boldsymbol{\theta}_t)\Big|\hat{\boldsymbol{\xi}}_{t-1}\right] = \mathbb{E}_{\boldsymbol{\xi}_t}\left[\frac{1}{b_t}\sum_{i=1}^{b_t}\nabla f_{\xi_{t,i}}(\boldsymbol{\theta}_t)\Big|\hat{\boldsymbol{\xi}}_{t-1}\right] = \frac{1}{b_t}\sum_{i=1}^{b_t}\mathbb{E}_{\xi_{t,i}}\left[\nabla f_{\xi_{t,i}}(\boldsymbol{\theta}_t)\Big|\hat{\boldsymbol{\xi}}_{t-1}\right],$$

which, together with Assumption 1(A2)(i) and the independence of $\boldsymbol{\xi}_t$ and $\boldsymbol{\xi}_{t-1}$, implies that

$$\mathbb{E}_{\boldsymbol{\xi}_t}\left[\nabla f_{B_t}(\boldsymbol{\theta}_t)\Big|\hat{\boldsymbol{\xi}}_{t-1}\right] = \frac{1}{b_t}\sum_{i=1}^{b_t}\nabla f(\boldsymbol{\theta}_t) = \nabla f(\boldsymbol{\theta}_t). \tag{17}$$

Assumption 1(A3), the independence of $b_t$ and $\boldsymbol{\xi}_t$, and (17) imply that

$$\begin{aligned}
\mathbb{V}_{\boldsymbol{\xi}_t}\left[\nabla f_{B_t}(\boldsymbol{\theta}_t)\Big|\hat{\boldsymbol{\xi}}_{t-1}\right] &= \mathbb{E}_{\boldsymbol{\xi}_t}\left[\|\nabla f_{B_t}(\boldsymbol{\theta}_t) - \nabla f(\boldsymbol{\theta}_t)\|^2\Big|\hat{\boldsymbol{\xi}}_{t-1}\right] \\
&= \mathbb{E}_{\boldsymbol{\xi}_t}\left[\left\|\frac{1}{b_t}\sum_{i=1}^{b_t}\nabla f_{\xi_{t,i}}(\boldsymbol{\theta}_t) - \nabla f(\boldsymbol{\theta}_t)\right\|^2\Big|\hat{\boldsymbol{\xi}}_{t-1}\right] \\
&= \frac{1}{b_t^2}\mathbb{E}_{\boldsymbol{\xi}_t}\left[\left\|\sum_{i=1}^{b_t}\left(\nabla f_{\xi_{t,i}}(\boldsymbol{\theta}_t) - \nabla f(\boldsymbol{\theta}_t)\right)\right\|^2\Big|\hat{\boldsymbol{\xi}}_{t-1}\right].
\end{aligned}$$

From the independence of $\xi_{t,i}$ and $\xi_{t,j}$ $(i \neq j)$ and Assumption 1(A2)(i), for all $i, j \in [b_t]$ such that $i \neq j$,

$$\begin{aligned}
&\mathbb{E}_{\xi_{t,i}}[\langle\nabla f_{\xi_{t,i}}(\boldsymbol{\theta}_t) - \nabla f(\boldsymbol{\theta}_t), \nabla f_{\xi_{t,j}}(\boldsymbol{\theta}_t) - \nabla f(\boldsymbol{\theta}_t)\rangle|\hat{\boldsymbol{\xi}}_{t-1}] \\
&= \langle\mathbb{E}_{\xi_{t,i}}[\nabla f_{\xi_{t,i}}(\boldsymbol{\theta}_t)|\hat{\boldsymbol{\xi}}_{t-1}] - \mathbb{E}_{\xi_{t,i}}[\nabla f(\boldsymbol{\theta}_t)|\hat{\boldsymbol{\xi}}_{t-1}], \nabla f_{\xi_{t,j}}(\boldsymbol{\theta}_t) - \nabla f(\boldsymbol{\theta}_t)\rangle \\
&= 0.
\end{aligned}$$

Hence, Assumption 1(A2)(ii) guarantees that

$$\mathbb{V}_{\boldsymbol{\xi}_t}\left[\nabla f_{B_t}(\boldsymbol{\theta})\Big|\hat{\boldsymbol{\xi}}_{t-1}\right] = \frac{1}{b_t^2}\sum_{i=1}^{b_t}\mathbb{E}_{\xi_{t,i}}\left[\|\nabla f_{\xi_{t,i}}(\boldsymbol{\theta}_t) - \nabla f(\boldsymbol{\theta}_t)\|^2\Big|\hat{\boldsymbol{\xi}}_{t-1}\right] \leq \frac{\sigma^2 b_t}{b_t^2} = \frac{\sigma^2}{b_t},$$

which completes the proof. $\qquad\square$

## C.2 Proof of Theorem 1

In this section, we prove Theorem 1 for the case of the NSHB algorithm, where the learning rate is denoted by $\eta_t$. The proof for the SHB algorithm, which uses the learning rate $\alpha_t$, follows an analogous argument. In particular, if we define $\eta_t = \frac{\alpha_t}{1-\beta}$, then the SHB update rule becomes equivalent to that of NSHB. Therefore, it is sufficient to prove the result for NSHB only.

*Proof.* From the $L$-smoothness of the function $f$, the descent lemma (Beck, 2017, Lemma 5.7) holds. That is,

$$f(\boldsymbol{\theta}_{t+1}) \leq f(\boldsymbol{\theta}_t) + \langle\nabla f(\boldsymbol{\theta}_t), \boldsymbol{\theta}_{t+1} - \boldsymbol{\theta}_t\rangle + \frac{L}{2}\|\boldsymbol{\theta}_{t+1} - \boldsymbol{\theta}_t\|^2.$$

Applying the update rule $\boldsymbol{\theta}_{t+1} = \boldsymbol{\theta}_t - \eta_t \boldsymbol{m}_t$ gives

$$f(\boldsymbol{\theta}_{t+1}) \leq f(\boldsymbol{\theta}_t) - \eta_t\langle\nabla f(\boldsymbol{\theta}_t), \boldsymbol{m}_t\rangle + \frac{L}{2}\eta_t^2\|\boldsymbol{m}_t\|^2. \tag{18}$$

By expanding $\boldsymbol{m}_t = \beta \boldsymbol{m}_{t-1} + (1 - \beta)\nabla f_{B_t}(\boldsymbol{\theta}_t)$, we obtain

$$\langle \nabla f(\boldsymbol{\theta}_t), \boldsymbol{m}_t \rangle = \beta \langle \nabla f(\boldsymbol{\theta}_t), \boldsymbol{m}_{t-1} \rangle + (1 - \beta)\langle \nabla f(\boldsymbol{\theta}_t), \nabla f_{B_t}(\boldsymbol{\theta}_t) \rangle$$

and

$$\begin{aligned}
\|\boldsymbol{m}_t\|^2 &= \|\beta \boldsymbol{m}_{t-1} + (1 - \beta)\nabla f_{B_t}(\boldsymbol{\theta}_t)\|^2 \\
&= \beta^2 \|\boldsymbol{m}_{t-1}\|^2 + 2\beta(1 - \beta)\langle \nabla f_{B_t}(\boldsymbol{\theta}_t), \boldsymbol{m}_{t-1} \rangle + (1 - \beta)^2 \|\nabla f_{B_t}(\boldsymbol{\theta}_t)\|^2.
\end{aligned}$$

By Proposition 1,

$$\begin{aligned}
\mathbb{E}_{\boldsymbol{\xi}_t}\left[\|\nabla f_{B_t}(\boldsymbol{\theta}_t)\|^2 \,|\hat{\boldsymbol{\xi}}_{t-1}\right] &= \mathbb{E}_{\boldsymbol{\xi}_t}\left[\|\nabla f_{B_t}(\boldsymbol{\theta}_t) - \nabla f(\boldsymbol{\theta}_t) + \nabla f(\boldsymbol{\theta}_t)\|^2 \,\Big|\hat{\boldsymbol{\xi}}_{t-1}\right] \\
&= \mathbb{E}_{\boldsymbol{\xi}_t}\left[\|\nabla f_{B_t}(\boldsymbol{\theta}_t) - \nabla f(\boldsymbol{\theta}_t)\|^2 \,\Big|\hat{\boldsymbol{\xi}}_{t-1}\right] + 2\mathbb{E}_{\boldsymbol{\xi}_t}\left[\langle \nabla f_{B_t}(\boldsymbol{\theta}_t) - \nabla f(\boldsymbol{\theta}_t), \nabla f(\boldsymbol{\theta}_t)\rangle\Big|\hat{\boldsymbol{\xi}}_{t-1}\right] \\
&\quad + \mathbb{E}_{\boldsymbol{\xi}_t}\left[\|\nabla f(\boldsymbol{\theta}_t)\|^2 \,\Big|\hat{\boldsymbol{\xi}}_{t-1}\right] \\
&\leq \frac{\sigma^2}{b_t} + \|\nabla f(\boldsymbol{\theta}_t)\|^2.
\end{aligned}$$

Hence, taking the expectation conditioned on $\boldsymbol{\xi}_{t-1} = \hat{\boldsymbol{\xi}}_{t-1}$, we have

$$\mathbb{E}_{\boldsymbol{\xi}_t}[\langle \nabla f(\boldsymbol{\theta}_t), \boldsymbol{m}_t \rangle|\hat{\boldsymbol{\xi}}_{t-1}] = \beta \langle \nabla f(\boldsymbol{\theta}_t), \boldsymbol{m}_{t-1} \rangle + (1 - \beta)\|\nabla f(\boldsymbol{\theta}_t)\|^2$$

and

$$\mathbb{E}_{\boldsymbol{\xi}_t}[\|\boldsymbol{m}_t\|^2|\hat{\boldsymbol{\xi}}_{t-1}] \leq \beta^2 \|\boldsymbol{m}_{t-1}\|^2 + 2\beta(1 - \beta)\langle \nabla f(\boldsymbol{\theta}_t), \boldsymbol{m}_{t-1} \rangle + (1 - \beta)^2 \left(\frac{\sigma^2}{b_t} + \|\nabla f(\boldsymbol{\theta}_t)\|^2\right).$$

Taking the total expectation, we get

$$\mathbb{E}[\langle \nabla f(\boldsymbol{\theta}_t), \boldsymbol{m}_t \rangle] = \beta \mathbb{E}[\langle \nabla f(\boldsymbol{\theta}_t), \boldsymbol{m}_{t-1} \rangle] + (1 - \beta)\mathbb{E}[\|\nabla f(\boldsymbol{\theta}_t)\|^2]$$

and

$$\mathbb{E}[\|\boldsymbol{m}_t\|^2] \leq \beta^2 \mathbb{E}[\|\boldsymbol{m}_{t-1}\|^2] + 2\beta(1 - \beta)\mathbb{E}[\langle \nabla f(\boldsymbol{\theta}_t), \boldsymbol{m}_{t-1} \rangle] + (1 - \beta)^2 \left(\frac{\sigma^2}{b_t} + \mathbb{E}[\|\nabla f(\boldsymbol{\theta}_t)\|^2]\right). \tag{19}$$

Therefore, taking the total expectation on both sides of (18),

$$\begin{aligned}
\mathbb{E}[f(\boldsymbol{\theta}_{t+1}) - f(\boldsymbol{\theta}_t)] &= -\eta_t \mathbb{E}[\langle \nabla f(\boldsymbol{\theta}_t), \boldsymbol{m}_t \rangle] + \frac{L}{2}\eta_t^2 \mathbb{E}[\|\boldsymbol{m}_t\|^2] \\
&\leq -\left\{(1 - \beta)\eta_t - \frac{L}{2}(1 - \beta)^2 \eta_t^2\right\} \mathbb{E}[\|\nabla f(\boldsymbol{\theta}_t)\|^2] \\
&\quad - \left\{\beta \eta_t - L\beta(1 - \beta)\eta_t^2\right\}\mathbb{E}[\langle \nabla f(\boldsymbol{\theta}_t), \boldsymbol{m}_{t-1} \rangle] + \frac{L}{2}\beta^2 \eta_t^2 \mathbb{E}[\|\boldsymbol{m}_{t-1}\|^2] \\
&\quad + \frac{L}{2}(1 - \beta)^2 \eta_t^2 \frac{\sigma^2}{b_t}.
\end{aligned} \tag{20}$$

Defining the Lyapunov function $\mathcal{L}_t$ as

$$\mathcal{L}_t := \begin{cases} f(\boldsymbol{\theta}_t), & t = 0, \\ f(\boldsymbol{\theta}_t) + A_{t-1}\|\boldsymbol{m}_{t-1}\|^2, & t > 0, \end{cases}$$

we will first consider the case $t > 0$. Here, the following equality holds:

$$\mathbb{E}[\mathcal{L}_{t+1} - \mathcal{L}_t] = \mathbb{E}[f(\boldsymbol{\theta}_{t+1}) - f(\boldsymbol{\theta}_t)] + A_t \mathbb{E}[\|\boldsymbol{m}_t\|^2] - A_{t-1}\mathbb{E}[\|\boldsymbol{m}_{t-1}\|^2]. \tag{21}$$

From (19),

$$
\begin{aligned}
A_t \mathbb{E}[\|\boldsymbol{m}_t\|^2] - A_{t-1}\mathbb{E}[\|\boldsymbol{m}_{t-1}\|^2] \leq & \ A_t(1-\beta)^2 \mathbb{E}[\|\nabla f(\boldsymbol{\theta}_t)\|^2] \\
& + 2A_t\beta(1-\beta)\mathbb{E}[\langle \nabla f(\boldsymbol{\theta}_t), \boldsymbol{m}_{t-1}\rangle] - (A_{t-1} - \beta^2 A_t)\mathbb{E}[\|\boldsymbol{m}_{t-1}\|^2] \\
& + A_t(1-\beta)^2 \frac{\sigma^2}{b_t}.
\end{aligned} \tag{22}
$$

Therefore, by combining (20) and (22), we obtain the following expression for (21):

$$
\begin{aligned}
\mathbb{E}[\mathcal{L}_{t+1} - \mathcal{L}_t] \leq & \left[ -\left\{ (1-\beta)\eta_t - \frac{L}{2}(1-\beta)^2\eta_t^2 \right\} + A_t(1-\beta)^2 \right] \mathbb{E}[\|\nabla f(\boldsymbol{\theta}_t)\|^2] \\
& + \left[ -\left\{ \beta\eta_t - L\beta(1-\beta)\eta_t^2 \right\} + 2A_t\beta(1-\beta) \right] \mathbb{E}[\langle \nabla f(\boldsymbol{\theta}_t), \boldsymbol{m}_{t-1}\rangle] \\
& + \left\{ \frac{L}{2}\beta^2\eta_t^2 - (A_{t-1} - \beta^2 A_t) \right\} \mathbb{E}[\|\boldsymbol{m}_{t-1}\|^2] \\
& + \left\{ \frac{L}{2}(1-\beta)^2\eta_t^2 + A_t(1-\beta)^2 \right\} \frac{\sigma^2}{b_t}.
\end{aligned} \tag{23}
$$

In order to eliminate the term $\mathbb{E}[\langle \nabla f(\boldsymbol{\theta}_t), \boldsymbol{m}_{t-1}\rangle]$, we choose $A_t$ such that

$$
A_t = \frac{\eta_t - L(1-\beta)\eta_t^2}{2(1-\beta)}.
$$

To ensure that $A_t \geq 0$, we require $\eta_t \leq \frac{1}{L(1-\beta)}$. Under this choice, (23) simplifies to

$$
\begin{aligned}
\mathbb{E}[\mathcal{L}_{t+1} - \mathcal{L}_t] \leq & -\frac{1}{2}(1-\beta)\eta_t \mathbb{E}[\|\nabla f(\boldsymbol{\theta}_t)\|^2] - \frac{1}{2}\left( \frac{\eta_{t-1} - \beta^2\eta_t}{1-\beta} - L\eta_{t-1}^2 \right) \mathbb{E}[\|\boldsymbol{m}_{t-1}\|^2] + \frac{1}{2}(1-\beta)\eta_t \frac{\sigma^2}{b_t} \\
\leq & -\frac{1}{2}(1-\beta)\eta_t \mathbb{E}[\|\nabla f(\boldsymbol{\theta}_t)\|^2] - \frac{1}{2}\left( \frac{1-c\beta^2}{1-\beta} - L\eta_{t-1} \right)\eta_{t-1}\mathbb{E}[\|\boldsymbol{m}_{t-1}\|^2] + \frac{1}{2}(1-\beta)\eta_t \frac{\sigma^2}{b_t} \\
\leq & -\frac{1}{2}(1-\beta)\eta_t \mathbb{E}[\|\nabla f(\boldsymbol{\theta}_t)\|^2] + \frac{1}{2}(1-\beta)\eta_t \frac{\sigma^2}{b_t}.
\end{aligned} \tag{24}
$$

The first inequality follows from the choice of the Lyapunov coefficient $A_t$. The second inequality uses the technical condition (2), i.e., $\frac{\eta_t}{\eta_{t-1}} \leq c$. The third inequality holds by assuming

$$
\eta_t \leq \frac{1-c\beta^2}{L(1-\beta)},
$$

which ensures that the coefficient of $\mathbb{E}[\|\boldsymbol{m}_{t-1}\|^2]$ is non-positive and therefore removable from the upper bound.

Furthermore, since $1 \leq c < \frac{1}{\beta^2}$, it follows that $\eta_t \leq \frac{1-c\beta^2}{L(1-\beta)} \leq \frac{1}{L(1-\beta)}$, which guarantees that the Lyapunov coefficient satisfies $A_t \geq 0$.

Next, in the case $t = 0$, $\boldsymbol{m}_{-1} = \boldsymbol{0}$, together with (19) and (20), leads to

$$
\begin{aligned}
\mathbb{E}[\mathcal{L}_1 - \mathcal{L}_0] = & \ \mathbb{E}[f(\boldsymbol{\theta}_1) - f(\boldsymbol{\theta}_0)] + A_0 \mathbb{E}[\|\boldsymbol{m}_0\|^2] \\
\leq & \left[ -\left\{ (1-\beta)\eta_0 - \frac{L}{2}(1-\beta)^2\eta_0^2 \right\} + A_0(1-\beta)^2 \right] \mathbb{E}[\|\nabla f(\boldsymbol{\theta}_0)\|^2] \\
& + \left\{ \frac{L}{2}(1-\beta)^2\eta_0^2 + A_0(1-\beta)^2 \right\} \frac{\sigma^2}{b_0} \\
= & -\frac{1}{2}(1-\beta)\eta_0 \mathbb{E}[\|\nabla f(\boldsymbol{\theta}_0)\|^2] + \frac{1}{2}(1-\beta)\eta_0 \frac{\sigma^2}{b_0}.
\end{aligned} \tag{25}
$$

Applying (25) to $t = 0$ and (24) to $1 \leq t \leq T - 1$ and then summing over $t = 0, \ldots, T - 1$ yields

$$\frac{1}{2}(1 - \beta) \sum_{t=0}^{T-1} \eta_t \mathbb{E}[\|\nabla f(\boldsymbol{\theta}_t)\|^2] \leq \mathbb{E}[\mathcal{L}_0 - \mathcal{L}_T] + \frac{1}{2}(1 - \beta)\sigma^2 \sum_{t=0}^{T-1} \frac{\eta_t}{b_t}. \tag{26}$$

From the definition of $\mathcal{L}_t$, we have

$$\begin{aligned}
\mathbb{E}[\mathcal{L}_0 - \mathcal{L}_T] &= \mathbb{E}[f(\boldsymbol{\theta}_0) - f(\boldsymbol{\theta}_T)] - A_{T-1}\mathbb{E}[\|\boldsymbol{m}_{T-1}\|^2] \\
&\leq \mathbb{E}[f(\boldsymbol{\theta}_0) - f(\boldsymbol{\theta}_T)] \\
&\leq f(\boldsymbol{\theta}_0) - f^\star.
\end{aligned}$$

The final inequality follows from the existence of the lower bound $f^\star$ of $f$. Therefore, (26) implies

$$\sum_{t=0}^{T-1} \eta_t \mathbb{E}[\|\nabla f(\boldsymbol{\theta}_t)\|^2] \leq \frac{2(f(\boldsymbol{\theta}_0) - f^\star)}{1 - \beta} + \sigma^2 \sum_{t=0}^{T-1} \frac{\eta_t}{b_t}.$$

Finally, since $\sum_{t=0}^{T-1} \eta_t > 0$, it follows that

$$\min_{0 \leq t \leq T-1} \mathbb{E}[\|\nabla f(\boldsymbol{\theta}_t)\|^2] \leq \frac{2(f(\boldsymbol{\theta}_0) - f^\star)}{1 - \beta} \frac{1}{\sum_{t=0}^{T-1} \eta_t} + \sigma^2 \frac{1}{\sum_{t=0}^{T-1} \eta_t} \sum_{t=0}^{T-1} \frac{\eta_t}{b_t}.$$

$\square$

### C.3 Proof of Corollary 1

*Proof.* We begin with the variance term $V_T$. Since the batch size $b$ is constant, we have:

$$V_T = \frac{1}{\sum_{t=0}^{T-1} \lambda_t} \sum_{t=0}^{T-1} \frac{\lambda_t}{b} = \frac{1}{b}.$$

We now proceed to analyze the term $B_T$ for different learning-rate schedules. The proof for the constant learning-rate case closely follows Theorem 3.1 in Umeda & Iiduka (2025). Let $\lambda_{\max} = \lambda$ denote the constant (maximum) learning rate.

[**Constant LR** (4)]     Under a constant learning rate $\lambda_t = \lambda$, we have:

$$B_T = \frac{1}{\sum_{t=0}^{T-1} \lambda} = \frac{1}{\lambda T}.$$

[**Diminishing LR** (5)]     Using the lower bound on a sum via integral approximation:

$$\sum_{t=0}^{T-1} \frac{1}{\sqrt{t+1}} \geq \int_0^T \frac{dt}{\sqrt{t+1}} = 2(\sqrt{T+1} - 1),$$

we obtain the following upper bound:

$$B_T = \frac{1}{\sum_{t=0}^{T-1} \frac{\lambda}{\sqrt{t+1}}} \leq \frac{1}{2\lambda(\sqrt{T+1} - 1)}.$$

[**Cosine LR** (6)]     We analyze the learning-rate schedule with a cosine decay:

$$\sum_{t=0}^{KE-1} \lambda_t = \lambda_{\min} KE + \frac{\lambda_{\max} - \lambda_{\min}}{2} KE + \frac{\lambda_{\max} - \lambda_{\min}}{2} \sum_{t=0}^{KE-1} \cos\left\lfloor \frac{t}{K} \right\rfloor \frac{\pi}{E}.$$

It can be shown that

$$\sum_{t=0}^{KE-1} \cos\left\lfloor \frac{t}{K} \right\rfloor \frac{\pi}{E} = K - 1 - \cos\pi = K, \tag{27}$$

so the total learning-rate sum becomes:

$$\sum_{t=0}^{KE-1} \lambda_t = \lambda_{\min}KE + \frac{\lambda_{\max} - \lambda_{\min}}{2}KE + \frac{\lambda_{\max} - \lambda_{\min}}{2}K$$

$$= \frac{1}{2}\{(\lambda_{\min} + \lambda_{\max})KE + (\lambda_{\max} - \lambda_{\min})K\}$$

$$\geq \frac{(\lambda_{\min} + \lambda_{\max})KE}{2}.$$

Finally, we obtain the upper bound:

$$B_T = \frac{1}{\sum_{t=0}^{KE-1} \lambda_t} \leq \frac{2}{(\lambda_{\min} + \lambda_{\max})KE}.$$

[**Polynomial LR** (7)]    Since $(1-x)^p$ is decreasing on $x \in [0,1)$, we use the inequality:

$$\int_0^1 (1-x)^p dx < \frac{1}{T}\sum_{t=0}^{T-1}\left(1 - \frac{t}{T}\right)^p,$$

which implies:

$$\sum_{t=0}^{T-1}\left(1 - \frac{t}{T}\right)^p > \frac{T}{p+1}.$$

Therefore, the total learning-rate sum satisfies:

$$\sum_{t=0}^{T-1} \lambda_t = (\lambda_{\max} - \lambda_{\min})\sum_{t=0}^{T-1}\left(1 - \frac{t}{T}\right)^p + \lambda_{\min}T$$

$$> \left(\frac{\lambda_{\max} - \lambda_{\min}}{p+1} + \lambda_{\min}\right)T = \frac{\lambda_{\max} + \lambda_{\min}p}{p+1}T.$$

Therefore, the bound for $B_T$ becomes:

$$B_T = \frac{1}{\sum_{t=0}^{T-1} \lambda_t} \leq \frac{p+1}{(\lambda_{\max} + \lambda_{\min}p)T}.$$

$\square$

## C.4   Proof of Corollary 2

*Proof.* We follow the approach outlined in the proof of Theorem A.1 in Umeda & Iiduka (2025). Let $M \in \mathbb{N}$ and define $T := \sum_{m=0}^{M-1} K_m E_m$, where $E_{\max} := \sup_{M \in \mathbb{N}} \sup_{0 \leq m \leq M} E_m < +\infty$, $K_{\max} := \sup_{M \in \mathbb{N}} \sup_{0 \leq m \leq M} K_m < +\infty$, $S_0 := \mathbb{N} \cap [0, K_0 E_0)$, and $S_m := \mathbb{N} \cap \left[\sum_{k=0}^{m-1} K_k E_k, \sum_{k=0}^{m} K_k E_k\right)$ $(m \in [M])$.

Consider the learning-rate sequence $\{b_t\}$ defined by the exponential growth schedule (10) with maximum parameter $\lambda_{\max} = \lambda$. By definition,

$$b_t = \delta^{m\left\lceil \frac{t}{\sum_{k=0}^{m} K_k E_k} \right\rceil} b_0,$$

where $\delta > 1$ and $b_0 > 0$.

For each $m$, we have

$$\sum_{t \in S_m} \frac{1}{b_t} = \sum_{t \in S_m} \frac{1}{\delta^{\left\lceil \frac{t}{\sum_{k=0}^m K_k E_k} \right\rceil} b_0} \leq \sum_{t \in S_m} \frac{1}{\delta^m b_0} = \frac{|S_m|}{\delta^m b_0} \leq \frac{K_{\max} E_{\max}}{\delta^m b_0},$$

where we used $\left\lceil \frac{t}{\sum_{k=0}^m K_k E_k} \right\rceil \geq 1$ and $|S_m| \leq K_{\max} E_{\max}$.

Summing over $m = 0, \ldots, M$, yields

$$\sum_{m=0}^{M-1} \sum_{t \in S_m} \frac{1}{b_t} \leq \frac{K_{\max} E_{\max}}{b_0} \sum_{m=0}^{M-1} \frac{1}{\delta^m} \leq \frac{\delta K_{\max} E_{\max}}{(\delta-1) b_0}. \tag{28}$$

[**Constant LR** (4)]     For the constant learning rate $\lambda_t = \lambda$, it holds that

$$V_T = \frac{1}{\sum_{t=0}^{T-1} \lambda} \sum_{t=0}^{T-1} \frac{\lambda}{b_t} = \frac{1}{T} \sum_{t=0}^{T-1} \frac{1}{b_t}.$$

Using inequality (28), we obtain

$$V_T \leq \frac{\delta K_{\max} E_{\max}}{(\delta-1) b_0 T}.$$

[**Diminishing LR** (5)]     For the diminishing learning rate $\lambda_t = \frac{\lambda}{\sqrt{t+1}}$, we have

$$V_T = \frac{1}{\sum_{t=0}^{T-1} \frac{\lambda}{\sqrt{t+1}}} \sum_{t=0}^{T-1} \frac{\lambda}{\sqrt{t+1} b_t} = \frac{1}{\sum_{t=0}^{T-1} \frac{1}{\sqrt{t+1}}} \sum_{t=0}^{T-1} \frac{1}{\sqrt{t+1} b_t} \leq \frac{1}{\sum_{t=0}^{T-1} \frac{1}{\sqrt{t+1}}} \sum_{t=0}^{T-1} \frac{1}{b_t}.$$

Since $\sum_{t=0}^{T-1} \frac{1}{\sqrt{t+1}} \geq 2(\sqrt{T+1} - 1)$, it follows that

$$V_T \leq \frac{\delta K_{\max} E_{\max}}{2(\delta-1) b_0 (\sqrt{T+1} - 1)}.$$

[**Cosine LR** (6)]     Suppose the learning rates satisfy $\lambda_{\min} \leq \lambda_t \leq \lambda_{\max}$. Then,

$$V_T = \frac{1}{\sum_{t=0}^{T-1} \lambda_t} \sum_{t=0}^{T-1} \frac{\lambda_t}{b_t} \leq \frac{\lambda_{\max}}{\sum_{t=0}^{T-1} \lambda_t} \sum_{t=0}^{T-1} \frac{1}{b_t}.$$

Applying bounds on $\sum_{t=0}^{T-1} \lambda_t \geq ((\lambda_{\min} + \lambda_{\max}) KE)/2$ under the cosine schedule yields

$$V_T \leq \frac{2 \delta \lambda_{\max} K_{\max} E_{\max}}{(\delta-1)(\lambda_{\min} + \lambda_{\max}) b_0 T}.$$

[**Polynomial LR** (7)]     For polynomially decaying learning rates with parameter $p > 0$ and $\lambda_{\min} \leq \lambda_t \leq \lambda_{\max}$, we similarly have

$$V_T \leq \frac{(p+1) \delta \lambda_{\max} K_{\max} E_{\max}}{(\delta-1)(p \lambda_{\min} + \lambda_{\max}) b_0 T}.$$

This completes the proof.                                                                                     $\square$

## C.5 Proof of Corollary 3

We follow the approach outlined in the proof of Theorem A.2 in Umeda & Iiduka (2025). Let $M \in \mathbb{N}$ and define $T := \sum_{m=0}^{M-1} K_m E_m$, where $E_{\max} := \sup_{M \in \mathbb{N}} \sup_{0 \le m \le M} E_m < +\infty$, $K_{\max} := \sup_{M \in \mathbb{N}} \sup_{0 \le m \le M} K_m < +\infty$, $S_0 := \mathbb{N} \cap [0, K_0 E_0)$, and $S_m := \mathbb{N} \cap \left[ \sum_{k=0}^{m-1} K_k E_k, \sum_{k=0}^{m} K_k E_k \right)$ $(m \in [M])$.

*Proof.* We have that

$$
\sum_{m=0}^{M-1} \sum_{t \in S_m} \lambda_t = \sum_{m=0}^{M-1} \sum_{t \in S_m} \gamma^{m \left\lceil \frac{t}{\sum_{k=0}^{m} K_k E_k} \right\rceil} \lambda_0 \ge \lambda_0 K_{\min} E_{\min} \sum_{m=0}^{M-1} \gamma^m
$$

$$
= \lambda_0 K_{\min} E_{\min} \frac{\gamma^M - 1}{\gamma - 1} > \frac{\lambda_0 K_{\min} E_{\min} \gamma^M}{\gamma^2} > \frac{\lambda_0 K_{\min} E_{\min} \gamma^M}{\delta^2}
$$

and

$$
\sum_{m=0}^{M-1} \sum_{t \in S_m} \frac{\lambda_t}{b_t} = \sum_{m=0}^{M-1} \sum_{t \in S_m} \frac{\gamma^{m \left\lceil \frac{t}{\sum_{k=0}^{m} K_k E_k} \right\rceil} \lambda_0}{\delta^{m \left\lceil \frac{t}{\sum_{k=0}^{m} K_k E_k} \right\rceil} b_0} \le K_{\max} E_{\max} \frac{\lambda_0}{b_0} \sum_{m=0}^{M-1} \frac{\gamma^m}{\delta^m}
$$

$$
\le K_{\max} E_{\max} \frac{\lambda_0}{b_0} \sum_{m=0}^{M-1} \left( \frac{\gamma}{\delta} \right)^m \le K_{\max} E_{\max} \frac{\lambda_0}{b_0} \frac{1}{1 - \hat{\gamma}},
$$

where $\hat{\gamma} = \frac{\gamma}{\delta} < 1$. Hence,

$$
B_T = \frac{1}{\sum_{t=0}^{T-1} \lambda_t} \le \frac{\delta^2}{\lambda_0 K_{\min} E_{\min} \gamma^M}
$$

and

$$
V_T = \frac{1}{\sum_{t=0}^{T-1} \lambda_t} \sum_{t=0}^{T-1} \frac{\lambda_t}{b_t} \le \frac{K_{\max} E_{\max} \lambda_0 \delta^2}{K_{\min} E_{\min} b_0 (1 - \hat{\gamma}) \gamma^M}.
$$

$\square$

## C.6 Proof of Corollary 4

Corollary 4 is directly obtained by applying the results established in Corollaries 2 and 3, and thus the proof is omitted.

# D Additional Experiments

The code used for the experiments is publicly available at `https://github.com/iiduka-researches/sgdm_lr_bs_schedule`

The details of the learning schedules (i)–(iv) used in the experiments are as follows.

**(i) Figure 5** The batch size is constant at 128, and the learning rate follows one of the schedules: Constant (4), Diminishing (5), Cosine (6), Polynomial (7) with $p = 2$, or Linear (7) with $p = 1$.

**(ii) Figure 6** The batch size doubles every 30 epochs, ranging from $2^3$ to $2^{12}$. The learning rate follows the same schedules as in (i). However, when the batch size is small, the number of steps per epoch becomes large, which causes the Polynomial and Linear schedules to decay rapidly in the initial phase.

**(iii) Figure 7** The batch size is the same as in (ii). The learning rate increases every 30 epochs by factors of 1.080, 1.196, and 1.292, starting from 0.1 and reaching 0.2, 0.5, and 1.0, respectively. To satisfy the condition in (2), we set $\beta = 0.87$ when $\eta_{\max} = 1.0$.

**(iv) Figure 8** The batch size is the same as in (ii). The learning rate increases every 3 epochs, starting from 0.1 and using the same multiplicative factors as in (iii). After this initial increase, it follows either the constant schedule (15) or the cosine schedule (16).

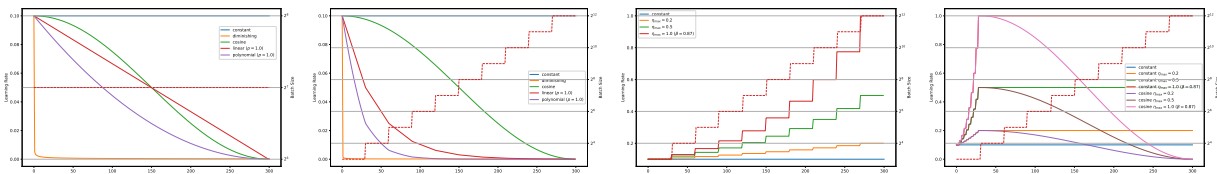

Figure 5: (i) Constant BS + decay LR

Figure 6: (ii) Increasing BS + decay LR

Figure 7: (iii) Increasing BS + increasing LR

Figure 8: (iv) Increasing BS + warmup LR

## D.1 Experimental Results for Normalized Stochastic Heavy Ball (NSHB)

We evaluated NSHB under the four schedule types illustrated in Figures 5–8. The detailed per-schedule plots are provided in Figures 9–12.

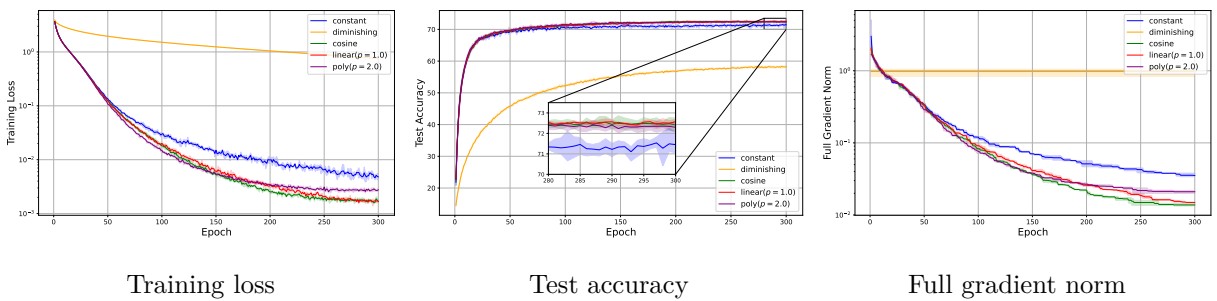

Training loss

Test accuracy

Full gradient norm

Figure 9: Results of NSHB under (i): constant batch size and decaying learning rate.

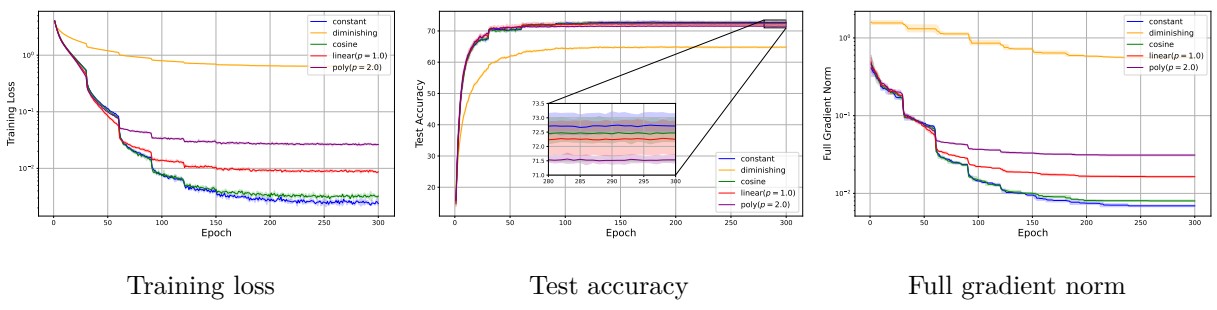

Training loss

Test accuracy

Full gradient norm

Figure 10: Results of NSHB under (ii): increasing batch size and decaying learning rate.

## D.2 Experimental Results for Stochastic Heavy Ball (SHB)

This section presents the experimental results of SHB under the four schedules (Figures 5–8). The SHB experiments included two learning rate settings, referred to as "high" and "low," which correspond approximately to the nominal learning rate of NSHB and one-tenth of that rate, respectively.

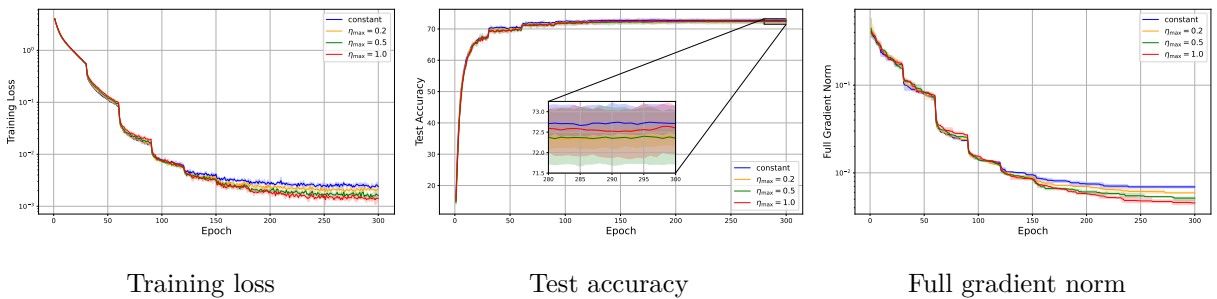

| Training loss | Test accuracy | Full gradient norm |

Figure 11: Results of NSHB under (iii): increasing batch size and increasing learning rate.

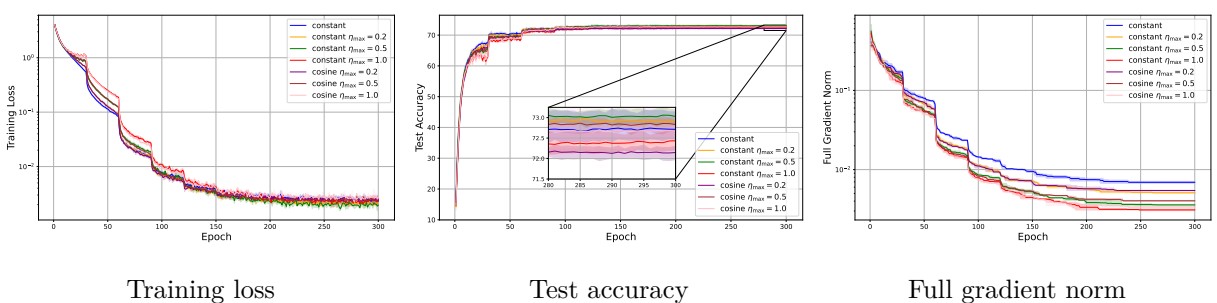

| Training loss | Test accuracy | Full gradient norm |

Figure 12: Results of NSHB under (iv): increasing batch size and warmup learning rate.

The use of a smaller learning rate for SHB is motivated by the allowable learning rates for NSHB and SHB, as specified in Theorem 1:

$$\eta_t \in \left[0, \frac{1 - c\beta^2}{L(1-\beta)}\right) \quad \text{(NSHB)}, \quad \alpha_t \in \left[0, \frac{1 - c\beta^2}{L}\right) \quad \text{(SHB)},$$

from which it follows that SHB requires a smaller learning rate than NSHB.

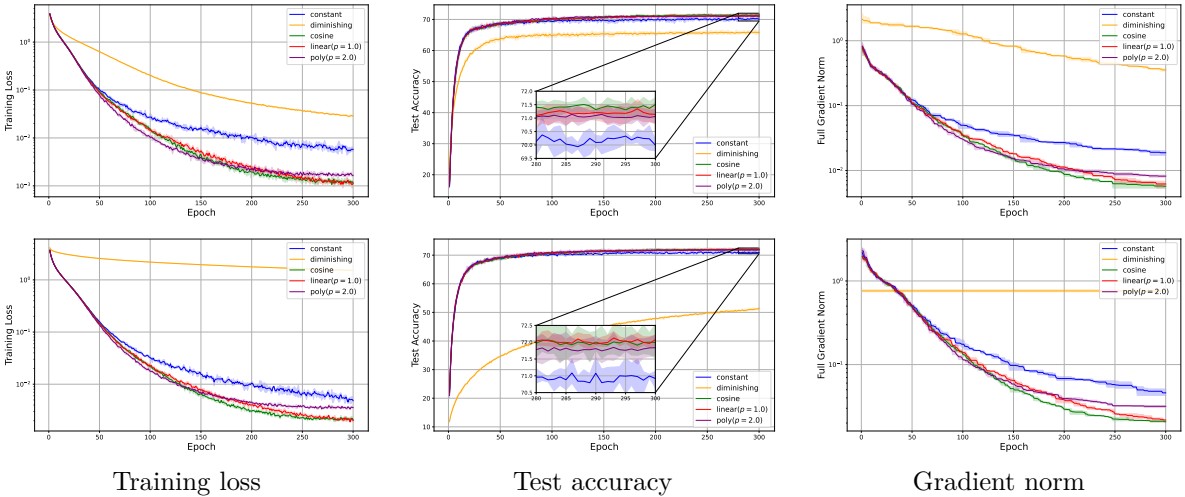

| Training loss | Test accuracy | Gradient norm |

Figure 13: Results of SHB under (i): constant batch size and decaying learning rate. Top row: learning rate = high (same as NSHB); bottom row: learning rate = low (one-tenth of NSHB).

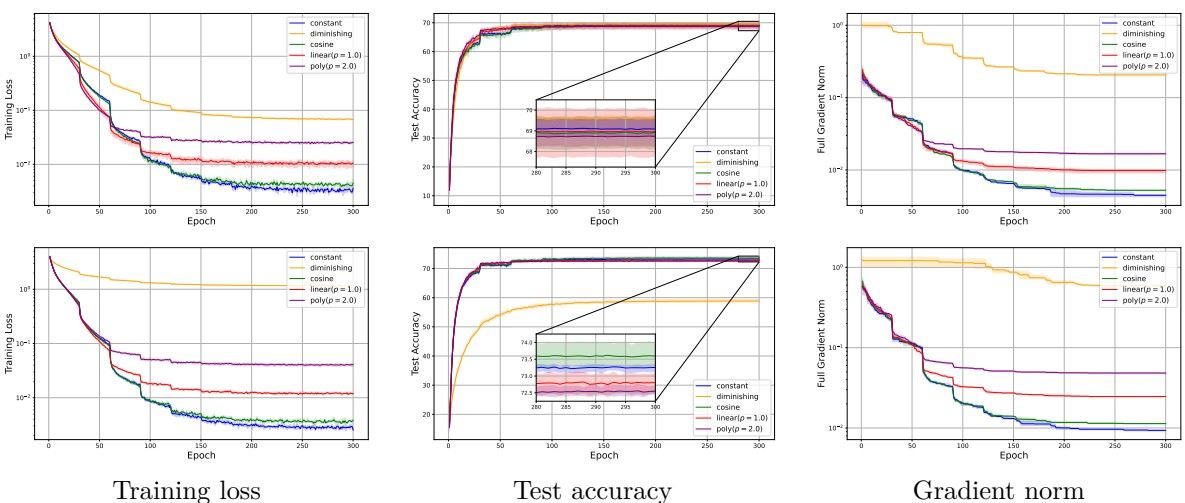

Figure 14: Results of SHB under (ii): increasing batch size and decaying learning rate.

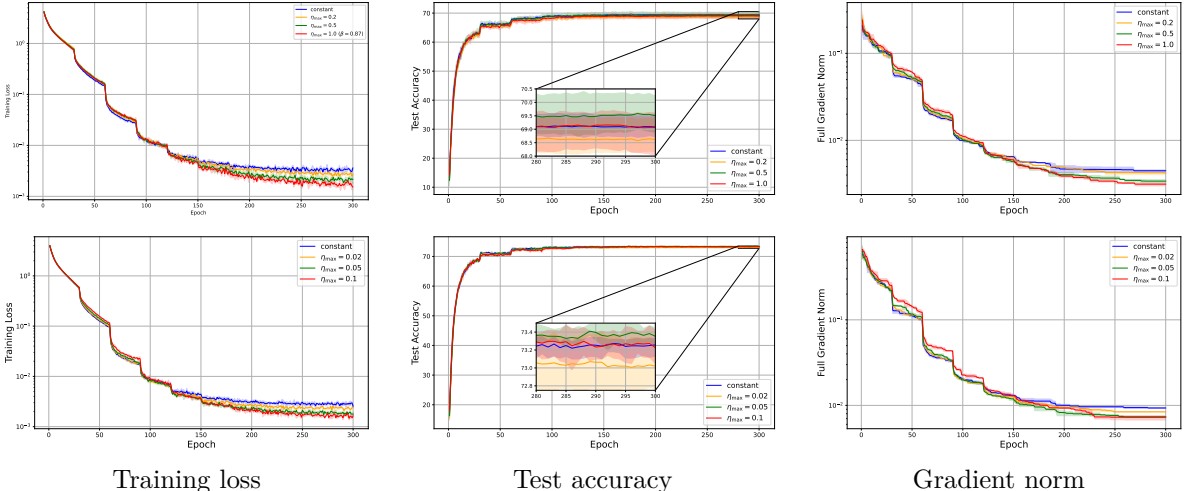

Figure 15: Results of SHB under (iii): increasing batch size and increasing learning rate.

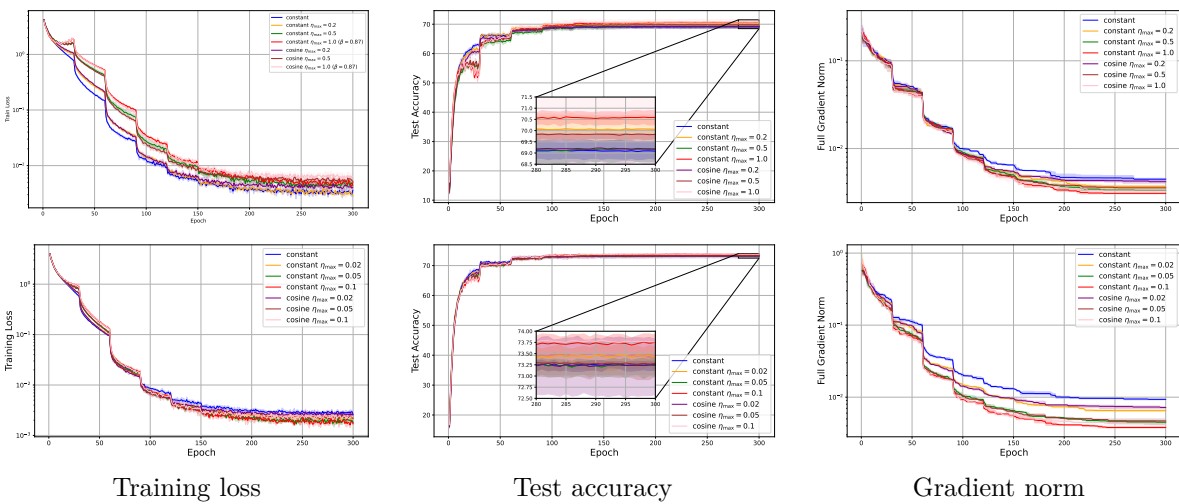

Figure 16: Results of SHB with (iv): increasing batch size and warmup learning rate.

### D.3 Experimental Results on ImageNet

We evaluated the empirical performance using ResNet-34 on the ImageNet dataset across 200 epochs. Figure 17 shows the trajectories of the training loss, test accuracy, and full gradient norm $\|\nabla f(\boldsymbol{\theta}_e)\|$ for the increasing batch size with a constant learning-rate schedule (schedule (ii)) and the proposed schedule with a warmup learning-rate schedule (schedule (iv)).

As illustrated in Figure 17, schedule (iv) achieves a lower training loss, a higher test accuracy, and a lower full gradient norm than schedule (ii) at the final epoch.

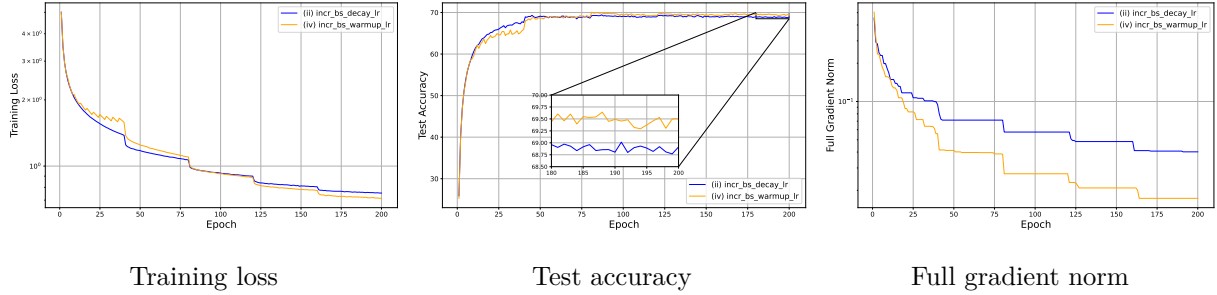

|                |               |                     |
|:--------------:|:-------------:|:-------------------:|
| Training loss  | Test accuracy | Full gradient norm  |

Figure 17: Empirical comparison between schedules (ii) and (iv) on ImageNet using ResNet-34.

## E   Extension to Dynamic Momentum Scheduling

Here, we provide the full details of the extension to dynamic momentum scheduling discussed in Section 5. We consider the case where the momentum coefficient $\beta_t$ is time-varying and monotonically non-increasing, i.e., $\beta_t \leq \beta_{t-1}$ for all $t$.

### E.1   Modified Lyapunov Function

For the dynamic momentum case, we modify the Lyapunov coefficient $A_t$ as follows:

$$A_t := \frac{\eta_t - L(1 - \beta_t)\eta_t^2}{2(1 - \beta_t)}.$$

Under the assumption that $\beta_t$ is monotonically non-increasing, the same descent argument as in the fixed $\beta$ case applies, and the learning rate condition takes the form:

$$\frac{1 - c\beta_t^2}{1 - \beta_{t-1}} - L\eta_{t-1} \geq 0.$$

### E.2   Convergence Bound

**Theorem 3** (Convergence bound for NSHB with dynamic momentum)**.** *Suppose Assumption 1 holds and $\{\beta_t\}$ is monotonically non-increasing. Then, for any $T \in \mathbb{N}$, the following bound holds:*

$$\min_{0 \leq t \leq T-1} \mathbb{E}\left[\|\nabla f(\boldsymbol{\theta}_t)\|^2\right] \leq \frac{2(f(\boldsymbol{\theta}_0) - f^\star)}{\sum_{t=0}^{T-1}(1 - \beta_t)\eta_t} + \frac{\sigma^2}{\sum_{t=0}^{T-1}(1 - \beta_t)\eta_t} \sum_{t=0}^{T-1} \frac{(1 - \beta_t)\eta_t}{b_t}.$$

Note that by substituting a specific schedule for $\beta_t$, one can derive concrete convergence rates.

**Proof Sketch.** We follow the same argument as in the proof of Theorem 1 (Appendix C.2), except for replacing the fixed $\beta$ with the time-varying $\beta_t$ in the Lyapunov coefficient $A_t$. Using the definition $\mathcal{L}_t :=$

$f(\boldsymbol{\theta}_t) + A_{t-1}\|\boldsymbol{m}_{t-1}\|^2$ and the modified coefficient $A_t$ above, the expected difference of the Lyapunov function satisfies:

$$
\begin{aligned}
&\mathbb{E}[\mathcal{L}_{t+1} - \mathcal{L}_t] \\
&= \mathbb{E}[f(\boldsymbol{\theta}_{t+1}) - f(\boldsymbol{\theta}_t)] + A_t\mathbb{E}[\|\boldsymbol{m}_t\|^2] - A_{t-1}\mathbb{E}[\|\boldsymbol{m}_{t-1}\|^2] \\
&\leq -\frac{1}{2}(1-\beta_t)\eta_t\mathbb{E}[\|\nabla f(\boldsymbol{\theta}_t)\|^2] - \frac{1}{2}\left(\frac{\eta_{t-1}}{1-\beta_{t-1}} - \frac{\beta_t^2\eta_t}{1-\beta_t} - L\eta_{t-1}^2\right)\mathbb{E}[\|\boldsymbol{m}_{t-1}\|^2] + \frac{1}{2}(1-\beta_t)\eta_t\frac{\sigma^2}{b_t} \\
&\leq -\frac{1}{2}(1-\beta_t)\eta_t\mathbb{E}[\|\nabla f(\boldsymbol{\theta}_t)\|^2] - \frac{1}{2}\left(\frac{1}{1-\beta_{t-1}} - \frac{c\beta_t^2}{1-\beta_t} - L\eta_{t-1}\right)\eta_{t-1}\mathbb{E}[\|\boldsymbol{m}_{t-1}\|^2] + \frac{1}{2}(1-\beta_t)\eta_t\frac{\sigma^2}{b_t} \\
&\leq -\frac{1}{2}(1-\beta_t)\eta_t\mathbb{E}[\|\nabla f(\boldsymbol{\theta}_t)\|^2] - \frac{1}{2}\left(\frac{1-c\beta_t^2}{1-\beta_{t-1}} - L\eta_{t-1}\right)\eta_{t-1}\mathbb{E}[\|\boldsymbol{m}_{t-1}\|^2] + \frac{1}{2}(1-\beta_t)\eta_t\frac{\sigma^2}{b_t} \\
&\leq -\frac{1}{2}(1-\beta_t)\eta_t\mathbb{E}[\|\nabla f(\boldsymbol{\theta}_t)\|^2] + \frac{1}{2}(1-\beta_t)\eta_t\frac{\sigma^2}{b_t}.
\end{aligned}
$$

The second inequality follows from the technical condition $\eta_t \leq c\eta_{t-1}$. The third inequality follows from the monotonicity assumption $\beta_t \leq \beta_{t-1}$. The final inequality holds by the learning rate condition $\frac{1-c\beta_t^2}{1-\beta_{t-1}} - L\eta_{t-1} \geq 0$, which ensures that the coefficient of $\mathbb{E}[\|\boldsymbol{m}_{t-1}\|^2]$ is non-positive. Summing over $t$ and rearranging terms, as in the proof of Theorem 1, yields the stated convergence bound.

