# OpenReview forum: "Accelerating SGDM via Learning Rate and Batch Size Schedules: A Lyapunov-Based Analysis"
_TMLR — Decision pending for TMLR_

### Review · Reviewer_pS1x · 2026-04-07

**Summary Of Contributions:**

This paper derives convergence results for SGD with heavy-ball momentum smooth, nonconvex problems under a rather general hyperparameter setup. The theoretical analysis of the paper is very general in the sense that it allows a broad class of learning-rate and batch-size schedules. While these hyperparameter schedules are still an important practical problem in deep learning, the general conclusion from the theory seem to be as expected: increasing the batch size or decreasing the learning rate reduces variance. The theory also reflects the finding that for increasing number of iterations, the optimal batch size should increase for optimal performance (or alternatively, that an increasing batch size schedule accelerates convergence).

Regarding the experiments, it seems that the main goal of the experiments is to validate the theoretical results and show that certain LR/batch size schedule perform better than others. Unfortunately, the paper does not extensively test how well the theory describes this interaction between training length and optimal batch size (see weaknesses below).

Main weaknesses of experiment section:

1) The lacking depth of the experiments, as only a single model & dataset is evaluated. Besides that, the experiments are run on CIFAR-100; it is unclear if many conclusions can be drawn from this that are relevant to more practical deep learning problems (LLM training, image classification on larger datasets, vision encoders, generative models, ...)

2) It seems that the peak learning rate is not independently tuned for each schedule. This will have strong confounding effects on the performance, as in general the optimal peak LR can be very different based on the shape of the schedule (for example, it is well know that for cosine the optimal LR is roughly twice as large as for constant with linear cooldown in the end). Figure 2 partially reveals this, as it shows that for example the gap between green and red is much bigger for the small learning rate than for the large one. Therefore, I expect that if the learning rates would have been tuned for each schedule independently, the results would look different.

**Additional Comments:**

Would it be possible to extend the analysis to the convex case (or other classes of functions, such as weakly convex)?

**Audience:**

Yes

**Audience Explanation:**

The theoretical results are interesting for optimization research, as they make previous results applicable in more general settings. This specifically applies to a wide range of step and batch size schedules, which are also important in practice.

**Claims And Evidence:**

No

**Claims Explanation:**

In general, my answer is yes. However, I think that the current experiment does not sufficiently back the claim of validating the theory. (fourth bullet point in contributions), due to the concerns mentioned above. If this can be clarified through revisions, I am happy to change my evaluation.

**Requested Changes:**

In general, I would recommend to strengthen the experimental evaluation by exploring different datasets or learning tasks, and by ruling out confounding factors through more extensive hyperparameter tuning (mainly, the learning rate).

Some further questions:

* The presentation with SHB and NSHB is slightly confusing because they are equivalent up to reparametrization of the learning rate (as the authors point out). It seems that the theoretical results for both methods are also not different when taking into account this reparametrization. Therefore, it might be much simpler to present the analysis for one of the two, and specify the analogous results for the other one in the appendix.

* The column "Function properties" in Table 1 is slightly confusing: first, it seems that the theory presented here is assuming Lipschitz smoothness of the gradients (which is not mentioned in the table). Second, the assumption of weak convexity for entry (ii) is explicitly mentioned, however this function class contains all Lipschitz-smooth functions as special case. Could the authors clarify this?

* An important reference for the analysis of SHB is https://proceedings.mlr.press/v134/sebbouh21a.html. Could the authors add this reference and a short discussion how their analysis is different?

* The plots from Figure 4 (batch size and LR schedules) should be part of the main text in order to compare the four compared settings. Further, I would recommend to not plot batch size and LR schedule in the same figure as it's done in Figure 4, as this makes the plot unnecessarily hard to read.

Minor:

* The first reference (Balles et al., has two different years listed).
* Proposition 1 seems like a standard result from the literature, and hence a reference would be useful. It seems that for Proposition 1, sampling with replacement is needed, but this could be easily extended to the case without replacement.

---

> ### Author Response · Authors · 2026-06-09
>
> Thank you for your careful and thorough review of our paper. Below we address each of your comments. All changes in the revised manuscript are highlighted in red.
>
> ---
>
> ## Experiments
>
> **Dataset and model diversity**
>
> We agree that evaluating only ResNet-18 on CIFAR-100 limits the depth of the experimental analysis. We have run experiments with ResNet-34 on ImageNet, and the results are provided in Appendix D.3. The full results will be included in the camera-ready version.
>
> **Learning rate tuning**
>
> We agree that the optimal peak learning rate for cosine annealing differs from that for a constant schedule. For the constant batch size setting, we confirmed that doubling the learning rate under cosine annealing improves performance. However, when using an increasing batch size, cosine annealing did not outperform the constant learning rate schedule. These results will also be included in the camera-ready version.
>
> **Placement of Figure 4**
>
> We have moved Figure 4 (learning rate and batch size schedule plots) into the main experimental section. We have also separated the learning rate and batch size schedules into individual figures for clarity.
>
> ---
>
> ## Presentation of SHB and NSHB
>
> We agree that presenting SHB and NSHB in parallel may confuse readers, given that they are equivalent up to a reparametrization of the learning rate. To improve readability, we have moved the SHB analysis to the Appendix and restructured the main text around the NSHB analysis (see Section 3).
>
> ---
>
> ## Table 1 (Function Properties column)
>
> [1] (Mai & Johansson, 2020) targets nonsmooth weakly convex functions and does not assume L-smoothness. We have updated the Function Properties column in Table 1 to explicitly indicate "L-smooth" for Ours, (i), and (ii) (second row), and changed (ii) (first row) to "weakly convex" only, so that the differences in assumptions across methods are clear (see Table 1).
>
> ---
>
> ## Additional reference
>
> We have added the reference suggested by the reviewer ([2] Sebbouh et al., 2021) and briefly discussed how it relates to our work (see Introduction).
>
> ---
>
> ## Minor
>
> - We have fixed the incorrect year in the Balles et al. reference.
> - We have added a reference for Proposition 1 and noted the assumption of sampling with replacement (see Appendix B.1).
>
> ---
>
> [1] Vien Mai, Mikael Johansson. "Convergence of a stochastic gradient method with momentum for non-smooth non-convex optimization". 2020.
>
> [2] Othmane Sebbouh, Robert M Gower, Aaron Defazio. "Almost sure convergence rates for Stochastic Gradient Descent and Stochastic Heavy Ball". 2021.

---

### Review · Reviewer_zkCU · 2026-04-09

**Summary Of Contributions:**

This paper studies SGD with momentum (SGDM) with learning-rate and batch-size scheduling, which is an important topic for training large models. The work extends the results of [Kamo & Iiduka, 2025] (studied SGDM with a constant learning rate) and [Umeda & Iiduka (2025)] (studied SGD without momentum with both increasing learning rate and batch size). Importantly, the paper studies practical learning rate schedules (e.g., LR warmup, cosine annealing) and batch size schedules (doubling batch size from time to time). The theoretical results are backed up by experiments on training Resnet18 model on CIFAR100 dataset.

**Additional Comments:**

- It is a bit surprising that decaying learning rate does not lead to convergence. Why is it so? Usually using $\sim\frac{1}{\sqrt{t}}$ stepsize is sufficient for convergence. Can the authors add a discussion on this aspect?

- Can the theory cover other batch size schedulers?

- Can the authors discuss the tightness of their results?

- Can the theory be extended to the convex case?

**Audience:**

Yes

**Audience Explanation:**

1. Batch size is an important parameter when training neural networks. Recent works demonstrated that it should not be treated as a constant, as it was before, but rather an additional parameter which should be tuned or adjusted during the training.

2. Theoretically demonstrating that learning rate and batch size schedulers lead to an improved convergence is an important result for the community, as it justifies the use of such schedulers in practice.

**Claims And Evidence:**

Yes

**Claims Explanation:**

# Strengths

1. The paper demonstrates that SGDM with a fixed batch size and learning rate converges to a neighborhood of the solution. Using learning rate and batch size schedulers fixes the issue. Using exponentially increasing learning rate and batch size even leads to a linear convergence instead of sublinear.

2. The authors provide empirical evidence that increasing batch size and learning rate is beneficial in practice, supporting the theoretical results of this work.

3. The proofs are clean and clear, it was easy to follow. I did not find mistakes there.


# Weaknesses

1. Why does the batch size only increase exponentially? The authors did not discuss why other batch size schedulers were not studied. Is it to make the calculations easier or for any other reason? Is exponential grow can be called optimal? The authors should also provide a discussion behind the tightness of the analysis to address this concern.

2. The experiments are not entirely fair, since neither the peak learning rate nor the initial batch size was tuned. Moreover, Adam exhibits unusual convergence behavior in the experiments, which may indicate that the chosen learning rates are suboptimal and should be tuned more carefully. Doing so would strengthen the paper’s empirical results. In addition, including a small GPT-2-style experiment, where batch-size scheduling is common practice, would further strengthen the empirical evaluation.

3. The discussion behind the Lyapunov function construction is somewhat limited. The authors did not provide sufficient amount of details to explain the construction. I also encourage to add comparison against other Lyapunov functions used in [1,2]. In general, the authors also should provide more details behind technical details in comparison with [1,2]. How large is the difference in the analysis? How difficult is to combine the results of [1,2]? Such a comparison would allow to show that the results of this work are not a trivial combination of [1] and [2].

[1] Kamo, Keisuke, and Hideaki Iiduka. "Increasing Batch Size Improves Convergence of Stochastic Gradient Descent with Momentum." arXiv preprint arXiv:2501.08883 (2025).

[2] Umeda, Hikaru, and Hideaki Iiduka. "Increasing both batch size and learning rate accelerates stochastic gradient descent." arXiv preprint arXiv:2409.08770 (2024).

**Requested Changes:**

- Please check previous sections.

- I encourage to polish the figures of this work. The legend and labels are small, it is hard to read them. The fonts should be significantly increased.

- Add details to Table 1:
    - what is $p_t$ and $d_t$, $\zeta$?;
    - add Lyapunov functions of [1,2] to the table.

---

> ### Author Response · Authors · 2026-06-09
>
> Thank you for your careful and thorough review of our paper. Below we address each of your comments. All changes in the revised manuscript are highlighted in red.
>
> ---
>
> ## Batch Size Scheduling
>
> The main motivation for studying only exponentially increasing batch sizes comes from [1] (Smith et al.). However, we acknowledge that exponential growth is not necessarily optimal. We will discuss the extension to other batch size schedulers and the tightness of our analysis as future work (see Conclusion).
>
> ---
>
> ## Experiments
>
> We are tuning the peak learning rate independently for each schedule. For the constant batch size setting, we confirmed that doubling the learning rate under cosine annealing improves performance. However, when using an increasing batch size, cosine annealing and other schedules did not outperform the constant learning rate. We are also tuning the learning rates for optimizers without learning rate scheduling, such as Adam. Additionally, we have run experiments with ResNet-34 on ImageNet, and the results are provided in Appendix D.3. The full results will be included in the camera-ready version.
>
> ---
>
> ## Lyapunov Function and Comparison with Prior Work
>
> **Table 1**
>
> Regarding the request to add the Lyapunov functions of [2] and [3] to Table 1: [2] (Umeda & Iiduka) analyzes SGD without momentum and therefore does not use a Lyapunov function. [3] (Kamo & Iiduka) uses the same form of Lyapunov function as entry (i) in our Table 1, and we have noted this in the table (see Table 1).
>
> **Lyapunov function construction**
>
> We will add more detail to the proof sketch of Theorem 1 to better explain the construction of the Lyapunov function (see Section 3).
>
> **Comparison with prior work**
>
> Since [3] extends [4] (Liu et al.) to the increasing batch size setting, we focus our comparison on [4] (see Section 3 and Section 5).
>
> [4] analyzes NSHB with a fixed batch size of 1 and a constant learning rate. The numerator of the variance term (our $V_T$) is $\eta^2$ in [4], whereas it is $\eta$ in our result. Similarly, setting $\beta = 0$ in our result to recover SGD gives a numerator of $\eta$ rather than $\eta^2$ as in [2]. This gap is the reason why convergence is not guaranteed under a fixed batch size with $\eta_t = O(1/\sqrt{t})$, and represents one limitation of our analysis.
>
> Extending [2] (Umeda & Iiduka, SGD) directly to SGDM is nontrivial because the cross-term $\mathbb{E}[\langle \nabla f(\theta_t), m_{t-1}\rangle]$ appears in the descent bound and its sign is indefinite. This is why we introduce a Lyapunov function.
>
> The variable transformation used in [4], $z_t = \frac{1}{1-\beta}\theta_t - \frac{\beta}{1-\beta}\theta_{t-1}$, gives $z_{t+1} - z_t = -\eta g_t$ under a constant learning rate, eliminating the momentum term. Under a dynamic learning rate $\eta_t$, however, this becomes
> $$z_{t+1} - z_t = -\eta_t g_t - \frac{\beta}{1-\beta}(\eta_t - \eta_{t-1})m_{t-1},$$
> introducing an additional residual term. This prevents the direct application of the Lyapunov analysis in [4] and is one of the key technical contributions of our work (see Section 5).
>
> ---
>
> ## Other
>
> - The reason why convergence is not guaranteed under a decaying learning rate is as discussed above. We have explicitly stated this as a limitation in the main text (see Section 3.3).
> - We have added definitions of $p_t$, $d_t$, and $\zeta$ to Table 1 (see Table 1).
> - We have increased the font size in all figures.
>
> ---
>
> [1] Samuel L. Smith, Pieter-Jan Kindermans, Chris Ying, Quoc V. Le. "Don't Decay the Learning Rate, Increase the Batch Size". 2018.
>
> [2] Hikaru Umeda, and Hideaki Iiduka. "Increasing both batch size and learning rate accelerates stochastic gradient descent". 2024.
>
> [3] Keisuke Kamo, Hideaki Iiduka. "Increasing Batch Size Improves Convergence of Stochastic Gradient Descent with Momentum". 2025.
>
> [4] Yanli Liu, Yuan Gao, and Wotao Yin. "An Improved Analysis of Stochastic Gradient Descent with Momentum". 2020.

---

### Review · Reviewer_hEuN · 2026-05-25

**Summary Of Contributions:**

the paper presents a unified convergence analysis of SGDM under dynamic learning rate and batch size schedules. the theoretical framework covers both Stochastic Heavy-Ball (SHB) and Normalized Stochastic Heavy-Ball (NSHB) algorithms under non-convex, $L$-smooth assumptions.

**main contributions of the work:**
1) the authors introduce a simplified Lyapunov function of the form: $L_t := f(\theta_t) + A_{t-1} \|m_{t-1}\|^2 \quad (t > 0)$. which is simpler than historical alternatives (e.g., Gadat et al., 2018; Liu et al., 2020) and easily accommodates dynamic step sizes.
2) unified analysis: general convergence bounds (Th 1) that map NSHB to SHB via the relation $\eta_t = \frac{\alpha_t}{1-\beta}$
3) analysed three regimes of BS and LR scheduling in Corollaries 1-3
4) empirical validation is quite promising for a theoretical paper: experiments on full CIFAR-100 using a ResNet-18 training, show that increasing BS combined with LR warmup yields the fastest convergence and best generalization. a good connection with practice!

**key strengths:**

**S1:** the proposed Lyapunov function is elegant. the parameter $A_t$ is chosen specifically to cancel out the difficult cross-term $\mathbb{E}[\langle \nabla f(\theta_t), m_{t-1} \rangle]$ in the descent lemma, leading to a highly readable proof

**S2:** unified framework for NSHB and SHB

**S3:** proving that exponentially growing BS paired with increased LR leads to exponential convergence $O(\gamma^{-M/2})$ is a strong positive result

**key weaknesses:**

**W1:** misleading claim on constant BS convergence: the paper claims that decaying LRs with a constant BS "does not guarantee convergence" and leaves a variance floor. this is a limitation of  specific Lyapunov analysis (due to the cancellation of the quadratic step-size noise term), not a property of SGDM. under standard step sizes, SGDM does converge to 0 with a constant batch size. so in my opinion, the paper presents a limitation of their specific Lyapunov proof technique as if it were a fundamental property of SGDM. it is better to clarify this claim few more times in the paper

**W2:** the condition $\frac{\lambda_{t+1}}{\lambda_t} \le c < \frac{1}{\beta^2}$ becomes highly restrictive when the momentum coefficient $\beta$ is close to 1 (eg, $\beta = 0.9$ or $0.99$). for large $\beta$, $c$ must be extremely close to 1, forcing the maximum allowable LR $\lambda_{\max}$ to be very small, which limits the practicality of the step-size bounds

**W3:** the paper focuses on optimization convergence to a stationary point. however, in deep learning, escaping poor local minima (especially sharp ones) is critical for generalization. the authors do not discuss how their schedules (esp. the exponential schedules where the LR-to-BS ratio decays to 0) affect these escape dynamics (look at Xie et al., ICLR 2021)

**W4:** as usual, about experiments: comparisons against RMSProp, Adam, and AdamW in Fig 3 use default learning rates (0.01 and 0.001) without tuning, i think tuning all methods a bit would enhance the empirical part of the work. even though it is not a crucial for the paper's claim as it is a theoretical work

**Audience:**

Yes

**Audience Explanation:**

this is an interesting paper with theoretical claims which are widely supported in deep learning practice (including but not limited to LLM pretraining). the entire analysis looks fine and elegant, and their topic in optimization theory overall is quite hot

**Broader Impact Concerns:**

no concerns on the ethical implications of the work. this is a theoretical paper

**Claims And Evidence:**

No

**Claims Explanation:**

with reservations regarding the interpretation of the constant BS variance floor, the paper is convincing and clear

**Requested Changes:**

**critical for acceptance:**

1) revise the text in Sec 3.3 and the Abstract. explicitly clarify that the variance floor $O(1/b)$ is a characteristic of the bound obtained via this specific Lyapunov function and not a proof that SGDM cannot converge under standard step-size decays with constant BS
2) add a detailed discussion in Sec 3.2 regarding the limits of the LR variation condition. explain how typical momentum settings ($\beta = 0.9$ or $0.99$) severely constrain the allowable rate of step-size increase ($c$) and force the maximum step size $\lambda_{\max}$ to be extremely small
3) clarify hyperparam tuning
4) elaborate in the main text on the mismatch between the warmup theory (which assumes synchronized BS and LR updates) and the experiments (which update them at different frequencies)

**strongly encouraged to strengthen the work:**

1) evaluate the schedules on a non-vision task (eg, training a small Transformer on Wikitext or Shakespeare) to show that the convergence speedup generalizes beyond CNN-like model. if you are going to do this, also cite a relevant literature
2) add a discussion section connecting your proposed BS and LR schedules to the escape dynamics of SGD. discuss how the exponential schedule (where $\lambda_t/b_t$ decays to 0) acts as simulated annealing to escape sharp local minima early in training. referencing Xie et al. 2021 *"A Diffusion Theory For Deep Learning Dynamics: Stochastic Gradient Descent Exponentially Favors Flat Minima"* would place the work in a broader and more impactful context
3) in the introduction, the statement that theoretical analyses of SGDM under dynamic schedules are extremely limited overlooks SDE and Langevin dynamics (SGLD) literature. works like Xie et al. already model SGD as diffusion processes to study the exact interplay of dynamic learning rates and batch sizes on landscape exploration and exit times. the authors should contextualize their optimization bounds within this broader SDE/diffusion literature
4) i encourage the authors to reference and discuss recent work studying the role of BS scaling in momentum methods, such as Islamov et al. *"On the Role of Batch Size in Stochastic Conditional Gradient Methods"* (ICML 2026). while that work focuses on conditional gradient methods (SCG/Scion) under a $\mu$-KL condition, it similarly demonstrates that static batch sizes suffer from a critical saturation threshold and proposes an adaptive schedule that increases BS during training. linking your SGDM convergence bounds to these parallel findings in constrained momentum optimization would provide a more unified perspective on BS scaling
5) the conclusion briefly mentions that extending this Lyapunov analysis to dynamic momentum scheduling $\beta_t$ is an open problem. provide a brief mathematical sketch in the appendix showing why a time-varying $\beta_t$ complicates the analysis. if it does not, you can just add it for free

---

> ### Author Response · Authors · 2026-06-09
>
> Thank you for your careful and thorough review of our paper. Below we address each of your comments. All changes in the revised manuscript are highlighted in red.
>
> ---
>
> ## Constant Batch Size and Convergence
>
> We agree with this point. The variance floor under a fixed batch size with a decaying learning rate is a limitation of our Lyapunov-based analysis, not a fundamental property of SGDM. Specifically, the numerator of the variance term is $\eta^2$ (or $\eta_t^2$) in [1] and [3], whereas it is $\eta_t$ in our result. This gap means that convergence is not guaranteed even with $\eta_t = O(1/\sqrt{t})$. We have revised the text in Section 3.3 and the Abstract to make this clear (see Section 3.3 and Abstract).
>
> ## Learning Rate Condition
> We agree that this is an important practical limitation and thank the reviewer for pointing it out. We have added a discussion to Section 3.2 addressing this constraint.
>
> ## Experiments
>
> We are tuning the peak learning rate independently for each schedule. For the constant batch size setting, we confirmed that doubling the learning rate under cosine annealing improves performance. However, when using an increasing batch size, cosine annealing and other schedules did not outperform the constant learning rate. We are also tuning the learning rates for optimizers without learning rate scheduling, such as Adam. Additionally, we have run experiments with ResNet-34 on ImageNet, and the results are provided in Appendix D.3. The full results will be included in the camera-ready version.
>
> ## Mismatch Between Theory and Warmup Experiment
>
> We agree that our theoretical analysis assumes synchronized updates of the learning rate and batch size, whereas the warmup experiment updates them at different frequencies. This choice is motivated by the critical batch size. If the batch size increase is stopped early, as with the learning rate warmup, most of the training would be conducted with a very large batch size, which may exceed the critical batch size and degrade performance. We note that the synchronized update setting is analyzed in Appendix A. We have added a discussion of this mismatch to the main text (see Section 4).
>
> ## Escape Dynamics
>
> We agree that in our exponential schedules, $\lambda_t/b_t \to 0$. Based on the results of [4] (Xie et al., ICLR 2021), this can be interpreted as promoting escape from sharp local minima early in training, while stabilizing convergence toward flat minima in later stages. We have added this perspective to the main text (see Introduction and Section 5).
>
> ## Additional References
>
> 1. We have revised the statement "theoretical analyses of SGDM under dynamic schedules are extremely limited" in the Introduction and added references to SDE and diffusion-based literature, including [4] (Xie et al., ICLR 2021) (see Introduction).
> 2. We have added [5] (Islamov et al., ICML 2026) to the related work. While their setting differs from ours in both the algorithm (SCG/Scion) and the assumption ($\mu$-KL condition), the shared finding that adaptively increasing the batch size accelerates convergence provides a complementary perspective (see Introduction).
>
> ## Dynamic Momentum Scheduling $\beta_t$
> We have found that the analysis can be extended to the case where $\beta_t$ is monotonically non-increasing. By defining the Lyapunov coefficient as
> $$A_t = \frac{\eta_t - L(1-\beta_t)\eta_t^2}{2(1-\beta_t)},$$
> the same argument as in the fixed $\beta$ case applies. The resulting convergence bound is
> $$\min_{0\le t\le T-1}\mathbb{E}[\|\nabla f(\theta_t)\|^2] \le \frac{2(f(\theta_0)-f^*)}{\sum_{t=0}^{T-1}(1-\beta_t)\eta_t} + \sigma^2\frac{\sum_{t=0}^{T-1}\frac{(1-\beta_t)\eta_t}{b_t}}{\sum_{t=0}^{T-1}(1-\beta_t)\eta_t}.$$
> We have added this result to the Appendix (see Section 5 and Appendix E).
>
> ---
>
> [1] Hikaru Umeda, and Hideaki Iiduka. "Increasing both batch size and learning rate accelerates stochastic gradient descent". 2024.
>
> [2] Keisuke Kamo, Hideaki Iiduka. "Increasing Batch Size Improves Convergence of Stochastic Gradient Descent with Momentum". 2025.
>
> [3] Yanli Liu, Yuan Gao, and Wotao Yin. "An Improved Analysis of Stochastic Gradient Descent with Momentum". 2020.
>
> [4] Zeke Xie, Issei Sato, Masashi Sugiyama. "A Diffusion Theory For Deep Learning Dynamics: Stochastic Gradient Descent Exponentially Favors Flat Minima". ICLR 2021.
>
> [5] Islamov et al. "On the Role of Batch Size in Stochastic Conditional Gradient Methods". ICML 2026.

---

### Decision · Action_Editor_rUiv · 2026-06-30

**Recommendation:** Accept as is

**Additional Comments:**

**Please incorporate the very last few comments from the reviewers, as noted in their recommendations.**

- Summarize contributions at the end of the paper for clarity.

- There appears to be a typo in this sentence: "However, extending such results to dynamic learning rate schedules remains challenging."

- On page 10, the text is duplicated: "On the Synchronization Mismatch between Theory and Experiments: On the Synchronization Mismatch between Theory and Experiments:"

- In Appendix E, the authors state "A detailed proof follows the same structure as Theorem 1 and is omitted here." since they claimed in their response that they added the result to Appendix E, they should at least provide a brief sketch of the proof to ensure completeness, rather than omitting it entirely

- In the references (page 14), the Balles et al. citation still lists the year 2017 twice

**Audience:**

Yes

**Audience Explanation:**

The results in the current forms are novel and interesting for the community. This is a very nice reference for people who want a handy analysis of schedulers (on LR and batch) in a general SGD setup, with useful experiments.

**Claims And Evidence:**

Yes

**Claims Explanation:**

The analysis in the paper was checked by reviewers and is based on standard techniques. The contribution here is to align these techniques with current strategies and design choices in modern deep learning optimization.

---

> ### Author Response · Authors · 2026-07-20
>
> We thank the Action Editor and the reviewers for their careful reading and valuable feedback throughout the review process.
>
> ---
>
> We have uploaded the camera-ready version incorporating the requested changes:
> - We added a brief proof sketch in Appendix E.
> - We added a summary of our contributions at the end of the paper (Conclusion).
> - We fixed the other minor issues pointed out by the reviewers, as well as a few additional typos we found during the revision process.